# A bacterial kinase phosphorylates OSK1 to suppress stomatal immunity in rice

Shanzhi Wang[1], Shuai Li[1], Jiyang Wang [1], Qian Li[1], Xiu-Fang Xin [2,3], Shuang Zhou[1], Yanping Wang[1], Dayong Li[4], Jiaqing Xu[1], Zhao-Qing Luo [5], Sheng Yang He[2,6] & Wenxian Sun [1,4✉]

The *Xanthomonas* outer protein C2 (XopC2) family of bacterial effectors is widely found in plant pathogens and *Legionella* species. However, the biochemical activity and host targets of these effectors remain enigmatic. Here we show that ectopic expression of XopC2 promotes jasmonate signaling and stomatal opening in transgenic rice plants, which are more susceptible to *Xanthomonas oryzae* pv. *oryzicola* infection. Guided by these phenotypes, we discover that XopC2 represents a family of atypical kinases that specifically phosphorylate OSK1, a universal adaptor protein of the Skp1-Cullin-F-box ubiquitin ligase complexes. Intriguingly, OSK1 phosphorylation at Ser[53] by XopC2 exclusively increases the binding affinity of OSK1 to the jasmonate receptor OsCOI1b, and specifically enhances the ubiquitination and degradation of JAZ transcription repressors and plant disease susceptibility through inhibiting stomatal immunity. These results define XopC2 as a prototypic member of a family of pathogenic effector kinases and highlight a smart molecular mechanism to activate jasmonate signaling.

[1] Department of Plant Pathology, the Ministry of Agriculture Key Laboratory of Pest Monitoring and Green Management, and Joint Laboratory for International Cooperation in Crop Molecular Breeding, Ministry of Education, China Agricultural University, Beijing, China. [2] DOE Plant Research Laboratory, Michigan State University, East Lansing, MI, USA. [3] National Key Laboratory of Plant Molecular Genetics, CAS Center for Excellence in Molecular Plant Sciences, Institute of Plant Physiology and Ecology, Chinese Academy of Sciences (CAS), CAS John Innes Centre of Excellence for Plant and Microbial Sciences (CEPAMS), Shanghai, China. [4] College of Plant Protection, Jilin Agricultural University, Changchun, Jilin, China. [5] Purdue Institute for Inflammation, Immunology and Infectious Disease and Department of Biological Sciences, Purdue University, West Lafayette, IN, USA. [6] Howard Hughes Medical Institute, Michigan State University, East Lansing, MI, USA. ✉email: wxs@cau.edu.cn

Many phytopathogenic bacteria inject effector proteins into host cells through the type III secretion system to inhibit plant defenses for successful infection[1–3]. The intense evolutionary arms race between the pathogen and its host, together with inter-kingdom horizontal gene transfer, generates remarkable sequence diversity of pathogen effectors[4], which makes it difficult to elucidate the biochemical functions of many pathogen effectors.

Elegant biochemical analyses combined with improved bioinformatics methods can facilitate the identification of novel functions associated with pathogen effectors, which provide insights into not only new pathogenicity mechanisms but also previously unrecognized aspects of host cell biology and signaling pathways[5]. For example, PSI-BLAST analysis revealed a putative mono-ADP-ribosyltransferase motif in the central region of members of the SidE effector family from *Legionella pneumophila*, which was subsequently found to catalyze a novel E1/E2-independent ubiquitination reaction[6]. The *Xanthomonas campestris* pv. *campestris* type III effector AvrAC was identified as a uridylyl transferase that modifies plant BIK1 and RIPK kinases, and consequently inhibits their kinase activity and downstream signaling[7]. The *Pseudomonas syringae* effector HopBF1 has been demonstrated to function as an atypical kinase that attacks the HSP90 chaperone of host cells[8]. Several other effector families have also been identified as protein kinases that target diverse host cellular processes[8–14]. These characterized effector kinases mainly belong to two classes. Class I effectors such as YpkA, XopAU, SteC, and LegK1 show a high sequence and structure similarity to eukaryotic kinases[12,15]; and class II effectors exemplified by NleH1, OspG, and HopBF1 harbor only basic kinase motifs and have lost several conserved subdomains found in canonical kinases[8,13,14].

*Xanthomonas* spp. cause many important diseases in a variety of plant species. *X. oryzae*, *X. campestris*, and *X. axonopodis* pathovars, for example, are among the top 10 important plant pathogenic bacteria[16]. *Xanthomonas* spp. secrete two classes of type III effectors, transcription activator-like (TAL) and non-TAL effectors, into host cells[17]. The TAL effectors are usually translocated into host cell nuclei and function as a unique family of transcription activators[18]. For example, *X. oryzae* PthXo1 activates the transcription of membrane-bound sugar transporter gene *Os8N3*, resulting in pumping out intracellular sugars into apoplastic spaces to feed bacteria[19,20]. AvrBs3 in *X. campestris* pv. *vesicatoria* targets a cell size regulator gene *upa20* to induce hypertrophy of plant mesophyll cells to promote infection[21].

The non-TAL effectors also play important roles in bacterial infection and disease development[22]. XopD is an active Ulp1-like cysteine protease that suppresses plant immunity by catalyzing the deSUMOylation and destabilization of transcription factor SlERF4 in tomato[23,24]. XopH is a 1-phytase that dephosphorylates myo-inositol-hexakisphosphate (InsP₆) to generate InsP₅ and interferes with plant hormone signaling[25]. Furthermore, several *Xanthomonas* effectors including XopK, XopL, and XopAE represent different types of ubiquitin E3 ligases[26–28]. Interestingly, XopAJ/AvrRxo1 functions as a NAD kinase, which phosphorylates NAD to produce 3'-NADP and thus suppresses ROS burst[29,30]. As a conventional serine/threonine kinase, XopAU in *X. euvesicatoria* manipulates MAPK signaling by phosphorylation and activation of MKK2[12]. Despite significant progress, the molecular mechanisms of the functions of most effector proteins in phytopathogenic bacteria remain unknown.

*X. oryzae* pv. *oryzicola* (*Xoc*) infects rice leaves through stomata and wounds and causes bacterial leaf streak, one of the most important bacterial diseases in rice[31]. Stomatal immunity greatly restricts bacterial infection at the very early infection stage[32]. As a countermeasure, phytopathogenic bacteria secrete effector

proteins and phytotoxins to suppress stomatal immunity. For instance, *P. syringae* generates the JA-mimicking phytotoxin coronatine and effector proteins, such as HopZ1 and HopX1, to activate JA signaling, thereby suppressing stomatal closure to facilitate bacterial entry of host tissues[33,34]. Several non-TAL effector genes, such as *avrBs2*, *hrpE3*, and *Xrp5*, are required for full virulence of *Xoc*[35–37]. However, little is known on the molecular mechanisms of how these effectors promote *Xoc* infection. In this study, we report that XopC2 in *Xoc* represents a family of core non-TAL type III effectors in *Xanthomonads*[35]. XopC2 homologs are widely distributed in other pathogenic bacterial species, such as *Acidovorax* and *Ralstonia* spp. We demonstrate that XopC2 functions as a novel type of kinase that phosphorylates OSK1, a universal adaptor protein of SCF complex, at Ser⁵³ residue. The phosphorylation of OSK1 at the specific site enhances the recruitment of OsCOI1b to the SCF complex and activates JA signaling.

## Results

**XopC2 defines a novel family of bacterial effector kinases**. PSI-BLAST analysis uncovered that XopC2 has homologs with high-level similarity in a wide range of phytopathogenic bacteria, including *A. citrulli* and *R. solanacearum*, and even in *Legionella* species (Supplementary Fig. 1). No known structural or functional domain was predicted in XopC2 and its homologs via SMART, Pfam, and Phyre[2] searches. However, a region in the carboxyl portion encompassing 391 to 417 amino-acid residues is highly conserved in these proteins revealed by sequence alignment and is predicted as a putative catalytic motif of protein kinases using HHpred (Fig. 1a and Supplementary Fig. 1). In addition, a P-loop-like motif featured with glycine-rich sequences and conserved lysine-serine/threonine (K-[S/T]) residues at the N-terminus might serve as a phosphate-binding motif (Supplementary Fig. 1). The conserved Lys¹⁴⁷, Asp³⁹¹, and Asp⁴¹³ residues of XopC2 are predicted to be the catalytic triad and the Asn³⁹⁶ residue most likely coordinates the second Mg²⁺ ion and is involved in phosphoryl transfer[38] (Fig. 1a, b and Supplementary Fig. 1). These characteristics prompted us to investigate whether XopC2 might function as a protein kinase via in vitro kinase assays. Indeed, purified XopC2 exhibited autophosphorylation (Fig. 1c). The mutated XopC2 proteins with Asp³⁹¹ and Asn³⁹⁶ residues replaced with Ala had a significantly reduced autophosphorylation activity (Fig. 1c). These results indicate that XopC2 is a functional protein kinase. Although XopC2 is not matched with any identified protein kinase from primary sequence alignment, the predicted secondary structure of XopC2 shows a similarity to the canonical protein kinase A (PKA). By contrast, XopC2 contains more α-helix subdomains in two central regions, one between subdomains III and IV and the other between subdomains V and VIa (Fig. 1b). The extra subdomains make the kinase domain of XopC2 (~470 amino acids) much longer than that of canonical kinases (250-300 amino acids). Phylogenetic analyses showed that XopC2 homologs form an independent cluster separated from any known protein kinase family (Supplementary Fig. 2). Thus, XopC2 represents the prototypic member of a family of protein kinases.

**XopC2-expressing rice plants exhibit increased disease susceptibility**. To investigate virulence functions of XopC2, 31 transgenic rice lines with constitutive expression of XopC2 driven by the CaMV 35S promoter (OE lines hereinafter) and 21, 24, and 27 transgenic lines with induced expression of XopC2 and the mutant variants XopC2^D391A and XopC2^N396A under dexamethasone (DEX)-inducible promoter (IE lines hereinafter), respectively, were generated through *Agrobacterium*-mediated

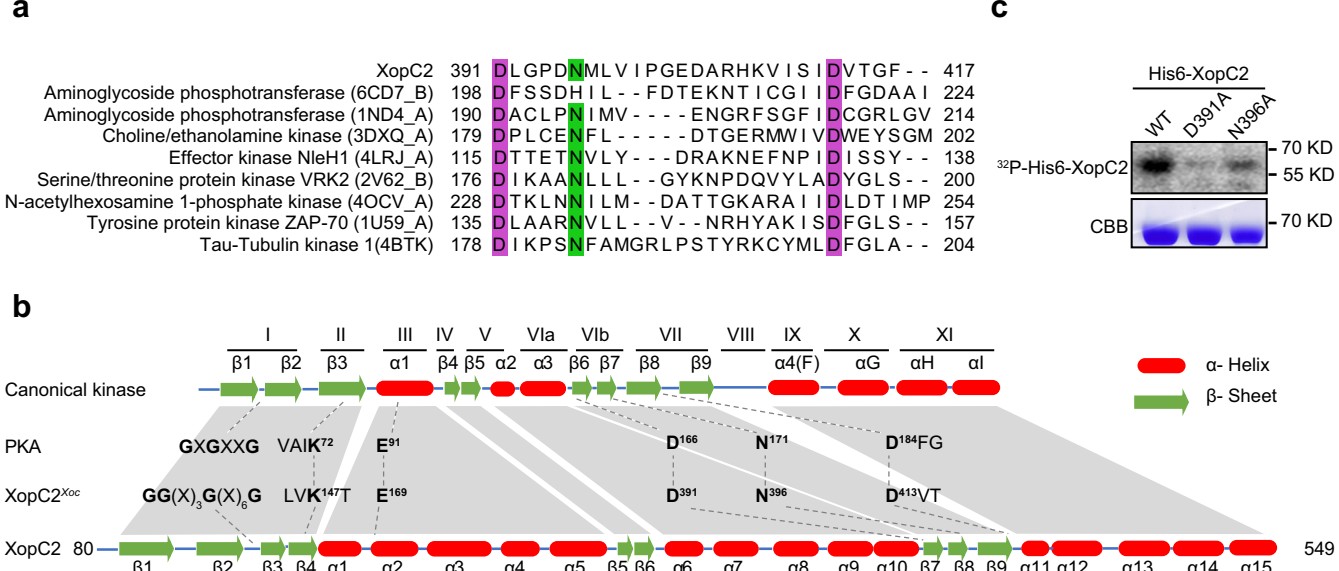

**Fig. 1 XopC2 represents a family of effector kinases. a** Sequence alignment of catalytic motifs in different families of protein kinases. Eight representative protein kinases with the highest scores in the output list from HHpred searches against the conserved motif (391–417 amino-acid residues) of XopC2 were aligned with ClustalX, and the conserved catalytic sites were highlighted in purple boxes. **b** Schematic diagrams of secondary structures of canonical kinases and XopC2. The top row showed the generally conserved secondary structure profile of canonical kinases[38]. Twelve subdomains and conserved structural characteristics were marked based on standard kinase annotations[70]. The bottom row showed the secondary structure profile of XopC2. The key conserved residues of canonical kinases and XopC2 were numbered based on the human canonical protein kinase A (PKA) and XopC2$^{Xoc}$, respectively, and were shown in the middle. **c** In vitro purified XopC2 was autophosphorylated, while the point mutations D391A and N396A reduced the autophosphorylation of XopC2 as revealed by in vitro kinase assays. Upper panel, His6-XopC2 phosphorylation was detected by autoradiography. Lower panel, the proteins were stained with Coomassie brilliant blue (CBB) as a loading control. The experiments were independently repeated 3 times with similar results.

transformation. Expression of XopC2-FLAG and its variants in these transgenic lines was detected by immunoblotting (Supplementary Fig. 3a, b). As compared with the wild-type plants, the independent homozygous OE-1, OE-10, IE-17, and IE-37 transgenic lines exhibited no alteration in growth and agronomic traits, including seedling and plant heights, leaf width, chlorophyll content, and hundred-grain weight (Supplementary Fig. 3).

After pressure infiltration with the wild-type *Xoc* RS105 and *ΔxopC2* mutant strains, the IE-17 transgenic line exhibited no significant difference in disease susceptibility regardless of mock and DEX treatments (Supplementary Fig. 4a). By contrast, the DEX-treated IE-17 transgenic line exhibited more disease lesions than the wild-type and mock-treated transgenic rice plants after spray inoculation with the *Xoc ΔxopC2* strain (Supplementary Fig. 4b). Consistent with disease symptoms, *in planta* bacterial population in the DEX-treated IE-17 transgenic line was significantly greater than those in the wild-type and mock-treated transgenic lines (Fig. 2a; Supplementary Fig. 4c). However, the DEX-treated IE-D391A-2 and IE-N396A-14 transgenic lines had equal *in-planta* bacterial population sizes to the wild-type and mock-treated transgenic plants (Fig. 2a). Collectively, these results suggest that XopC2 contributes to the initial steps of natural infection of *Xoc* in a kinase activity-dependent way.

In plants, stomatal immunity prevents phytopathogenic bacteria from entering host tissues as a barrier to initial infection[32]. To investigate whether XopC2 disarms stomatal immunity, stomatal conductance (Gs) was measured for rice leaves after spray inoculation with *ΔxopC2*. The DEX-treated IE-17 and IE-37 transgenic lines had significantly higher Gs than the wild-type and mock-treated IE lines after *ΔxopC2* inoculation, indicating that DEX-induced XopC2 expression compromises stomatal closure (Fig. 2b; Supplementary Fig. 4d). The wild-type,

IE-D391A-2, and IE-N396A-14 seedlings showed no significant difference in Gs regardless of mock and DEX treatments, although DEX-treated IE-N396A-14 seedlings had a slightly higher Gs than mock-treated seedlings after *ΔxopC2* infection (Fig. 2b). In addition, time course assays showed that the Gs of rice leaves was gradually decreased after *Xoc* infection. Compared with *ΔxopC2* and C-*ΔxopC2$^{D391A}$* challenges, rice leaves showed significantly higher Gs at 24 h after RS105 and C-*ΔxopC2* inoculation and thereafter (Supplementary Fig. 4e). Furthermore, the OE-1 and OE-10 lines showed higher Gs than the wild-type plants after *ΔxopC2* inoculation, while the wild-type and OE transgenic lines had similar Gs after RS105 infection (Supplementary Fig. 4f). These data indicate that XopC2, but not the catalytic mutants, suppresses stomatal closure triggered by *Xoc* infection.

**XopC2 promotes OsJAZ degradation and JA signaling.** Stomatal defense is often compromised when jasmonic acid (JA) signaling is activated in the context of pathogen infections[32]. To investigate this possibility, the expression of the JA-responsive marker genes including *OsLOX2* and *OsJAZ8* was analyzed in the *xopC2*-transgenic plants. *OsLOX2* and *OsJAZ8* expression was induced in the wild-type plants by exogenous methyl jasmonate (MeJA), but not by DEX treatment. By contrast, MeJA-induced expression of *OsLOX2* and *OsJAZ8* in IE-17 and IE-37 transgenic seedlings was significantly enhanced after DEX treatment compared with that in mock-treated transgenic seedlings (Fig. 3a, b; Supplementary Fig. 5a, b). However, MeJA-induced expression of *OsLOX2* and *OsJAZ8* was no longer enhanced in the DEX-treated IE-D391A-2 line, while was significantly promoted in the DEX-induced IE-N396A-14 line. Besides, JA-induced leaf senescence in the IE-17 and IE-37 lines was also promoted by DEX treatment

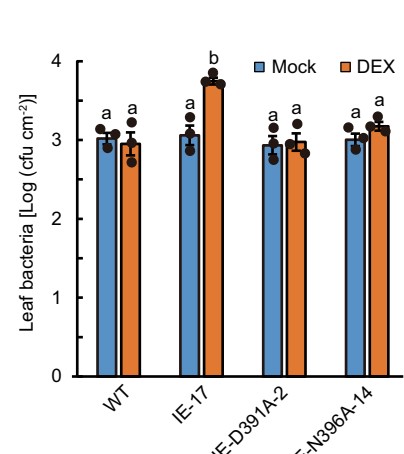 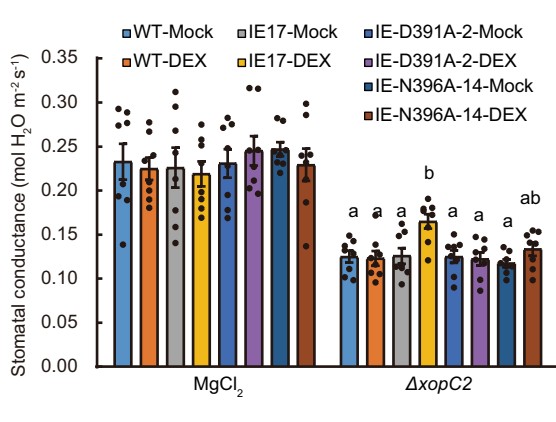

**Fig. 2 XopC2 suppresses rice stomatal defense to promote *X. oryzae* pv. *oryzicola* infection. a** Bacterial population sizes in the *ΔxopC2*-inoculated leaves of the wild-type, IE-17, IE-D391A-2, and IE-N396A-14 transgenic seedlings. The 3-week-old seedlings were treated with mock and DEX followed by spraying with *Xoc ΔxopC2* after 24 h. Bacterial populations were determined at 4 days post-inoculation (dpi). Data are shown as means ± standard error (SE) ($n = 3$ technical replicates per measurement). The letters (a, b) indicate a statistically significant difference in bacterial population sizes as revealed by one-way ANOVA, Tukey's honest significance test. **b** Stomatal conductance in the leaves of the wild-type, IE-17, IE-D391A-2, and IE-N396A-14 transgenic lines after challenging with *ΔxopC2*. The 3-week-old seedlings were pretreated with DEX and mock followed by spray inoculation with *ΔxopC2* and mock control. Stomatal conductance was measured at 2 dpi. Data are shown as means ± SE ($n = 8$ technical replicates per measurement). These experiments were independently repeated 3 times with similar results in **a** and **b**. The letters (a, b) indicate a statistically significant difference in stomatal conductance of the *xopC2* transgenic IE-17 seedlings between mock and DEX treatments as revealed by one-way ANOVA, Tukey's honest significance test.

compared with mock treatment (Supplementary Fig. 5c). In contrast, the SA and JA contents were not significantly different in mock-treated and DEX-treated wild-type and IE-17 lines (Supplementary Fig. 5d, e). In addition, either expression of *OsICS1* and *OsPAL1* (the SA-biosynthesis genes) or induced expression of *OsIAA9* and *D10* by auxin and strigolactone, respectively, was not altered in the IE transgenic lines after DEX treatment (Supplementary Fig. 5f–i). These results indicate that ectopically expressed XopC2 specifically promotes JA signaling in rice.

Jasmonate activates JA signaling by promoting JAZ degradation[39]. To investigate whether XopC2 induces degradation of OsJAZ proteins, HA-tagged OsJAZ7, OsJAZ9, OsJAZ12, and OsJAZ13 proteins were transiently expressed in rice protoplasts isolated from *xopC2* transgenic seedlings. The accumulation of all examined OsJAZs-HA was dramatically reduced in the DEX-treated transfected protoplasts compared with that in mock-treated ones as revealed by immunoblotting (Fig. 3c and Supplementary Fig. 6a). OsJAZ9-HA accumulation was not altered by DEX treatment per se when the protein was transiently expressed in the protoplasts prepared from wild-type seedlings (Supplementary Fig. 6b). In addition, MG132, a proteasome inhibitor, largely prevented XopC2-promoted OsJAZ degradation (Fig. 3c and Supplementary Fig. 6a). Collectively, these findings indicate that XopC2 promotes the 26S proteasome-mediated degradation of OsJAZs.

To examine OsJAZ degradation during *Xoc* infection, we generated the OsJAZ9-HA-NE and OsJAZ9-HA-OE transgenic rice plants expressing OsJAZ9-HA driven by the native and CaMV 35S promoters, respectively, (Supplementary Fig. 7a) and determined the stability of OsJAZ9-HA in these plants after inoculation of the wild-type, *ΔxopC2* or complemented *Xoc* strains. Spray inoculation of the wild-type strain caused evident degradation of OsJAZ9-HA in these transgenic seedlings, while OsJAZ9-HA was stable in the *ΔxopC2*-inoculated transgenic seedlings (Fig. 3d and Supplementary Fig. 7b). Interestingly, the plasmid-borne *xopC2* gene restored the ability of the C-*ΔxopC2*

complementation strain to promote OsJAZ9-HA degradation during infection, whereas the mutated *xopC2* gene encoding XopC2[D391A] did not (Fig. 3d and Supplementary Fig. 7b). Collectively, these results indicate that XopC2 secreted by *Xoc* promotes JAZ protein degradation during infection.

**XopC2 enhances OsJAZ9 ubiquitination in vitro.** JAZ proteins are ubiquitylated by the Skp1-Cullin-F-box type E3 ubiquitin ligase complex SCF[COI1] and are then subjected to degradation via the 26S proteasome[39]. To investigate whether XopC2 enhances ubiquitination of OsJAZ proteins, a semi-in vitro ubiquitination assay was performed. In this assay, His6-OsJAZ9 was first expressed in *E. coli* and then bound to nickel-agarose beads. His6-OsJAZ9-bound beads were then incubated with HA-ubiquitin and total cell extracts isolated from rice seedlings in the ATP-containing ubiquitination buffer. Immunoblotting with anti-HA-HRP showed that His6-OsJAZ9 bound to beads was ubiquitylated, while no ubiquitination signal was detected in the absence of total rice protein extracts (Fig. 3e). The ubiquitination of His6-OsJAZ9 was enhanced when GST-XopC2 was included in the reaction (Fig. 3e).

We next refined the semi-in vitro ubiquitination assay. The SCF[OsCOI1b] complex was immunoprecipitated from rice protoplasts expressing OsCOI1b-FLAG with anti-FLAG M2 affinity gel and was then incubated with E1, E2, ubiquitin, and His6-OsJAZ9 for ubiquitination assays. This assay showed that OsJAZ9 ubiquitination was significantly enhanced in the presence of GST-XopC2, whereas both GST-XopC2[D391A] and GST-XopC2[N396A] largely lost the ability to enhance His6-OsJAZ9 ubiquitination (Fig. 3f). The results suggest that the kinase activity of XopC2 is critical for its ability to promote JAZ ubiquitination.

**XopC2 interacts with and phosphorylates the adaptor protein OSK1 in SCF[COI1b].** To elucidate how XopC2, as a protein kinase, promotes JAZ ubiquitination, we investigated whether XopC2 phosphorylates specific component(s) of the SCF[COI1b] complex.

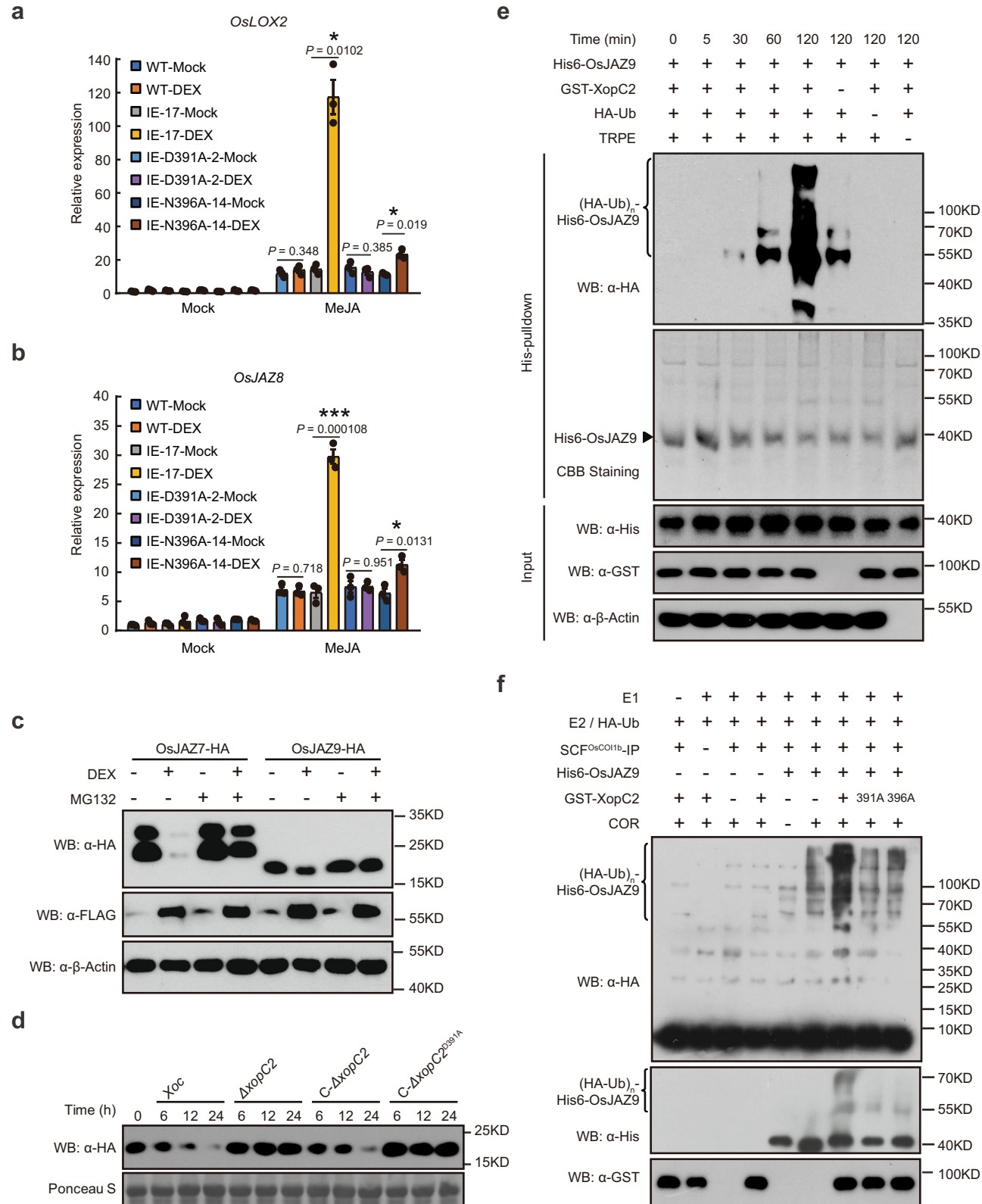

In vitro kinase assays showed that OSK1, but not other components including OsJAZ9, HA-ubiquitin, OsUBA1 (E1), UBCH5α (E2), OsCullin1a, OsRBX1, or OsCOI1b, was phosphorylated by XopC2 (Fig. 4a and Supplementary Fig. 8a). Subsequently, liquid chromatography-tandem mass spectrometry (LC-MS/MS)

revealed that three amino-acid residues including Thr[32], Ser[53], and Ser[92] in OSK1 were phosphorylated by XopC2 in vitro (Supplementary Fig. 8b). Based on structural modeling, the three residues together with seven other residues including Ser[15], Thr[62], Ser[83], Thr[133], Thr[142], Thr[149], and Thr[157] are predicted to reside

**Fig. 3 XopC2 enhances JA signaling by promoting ubiquitination and degradation of JAZ proteins. a, b** MeJA-induced expression of the JA-responsive genes *OsLOX2* (**a**) and *OsJAZ8* (**b**) was enhanced in XopC2-expressing rice seedlings. The wild-type, IE-17, IE-D391A-2, and IE-N396A-14 transgenic seedlings were treated with DEX or mock solution for 24 h followed by MeJA application (50 μM). Gene expression was detected by qRT-PCR using *OsActin* as an internal reference gene. Data are presented as means ± SE (*n* = 3 replicates per measurement). Asterisk (*) indicates a statistically significant difference in relative gene expression between mock and DEX treatments (two-sided *t*-test; *, *P* < 0.05; **, *P* < 0.01; ***, *P* < 0.001). **c** The accumulation of OsJAZs-HA was greatly reduced in rice protoplasts conditionally expressing XopC2-FLAG. OsJAZ7-HA and OsJAZ9-HA were transiently expressed in IE-17 transgenic rice protoplasts after mock and DEX treatments for 12 h. Western blotting was performed to detect OsJAZs-HA, XopC2-FLAG, and β-OsActin (as a protein loading control). MG132, a proteasome inhibitor. **d** OsJAZ9-HA was rapidly degraded during *Xoc* infection but remained relatively stable during Δ*xopC2* infection. Three-week-old transgenic seedlings expressing OsJAZ9-HA driven by its native promoter were sprayed with the wild-type, Δ*xopC2*, and Δ*xopC2* strains complemented with a wild-type copy of *xopC2* (C-Δ*xopC2*) or a kinase-defective copy (C-Δ*xopC2*^D391A^). OsJAZ9-HA was detected by immunoblotting at the indicated timepoints post-inoculation. **e** OsJAZ9 ubiquitination was enhanced in the presence of XopC2 in a semi-in vitro assay. After incubation with His6-OsJAZ9, NTA beads were divided equally and were mixed with GST-XopC2, HA-ubiquitin (Ub), and total rice protein extracts (TRPE) as indicated in the ubiquitination buffer. OsJAZ9 ubiquitination was detected at the indicated timepoints by immunoblotting with an anti-HA-HRP antibody. The protein loading was shown by CBB staining. **f** OsJAZ9 ubiquitination was enhanced in the presence of GST-XopC2 revealed by a refined semi-in vitro assay. The SCF^OsCOI1b^ complex was immunoprecipitated with anti-FLAG M2 affinity beads from the extract of rice protoplasts co-expressing OsCOI1b-FLAG, OsCullin1a-HA, OSK1-HA, and OsRBX1-HA. The SCF^OsCOI1b^ complex was then incubated with human UBE1 (E1), UBCH5α (E2), HA-ubiquitin (Ub), His6-OsJAZ9, coronatine (COR), and GST-XopC2/XopC2^D391A^/XopC2^N396A^ in the ATP-containing reaction buffer. OsJAZ9 ubiquitination was detected by immunoblotting with anti-HA-HRP and anti-His-HRP antibodies. The experiments were independently repeated 3 times with similar results in the panels **a–f**.

on the surface of OSK1 3-D structure and are candidate phosphosites (Supplementary Fig. 8c). To confirm the phosphorylation residues in OSK1 by XopC2, ten OSK1 variants with individual residues mutated to alanine were generated. The OSK1^T32A^, OSK1^S53A^, OSK1^S92A^, and OSK1^T149A^ variant proteins exhibited significantly reduced phosphorylation when they were incubated with XopC2 in vitro kinase assays (Supplementary Fig. 8d). These results indicate that these four residues are likely the major OSK1 phosphorylation sites by XopC2.

To detect whether XopC2 phosphorylates OSK1 during *Xoc* infection, we generated the transgenic rice plants expressing OSK1-FLAG (Supplementary Fig. 9a). The transgenic plants were inoculated with the wild-type, Δ*xopC2*, and *xopC2*-complemented strains. Total cell extracts were isolated from the inoculated leaves at 48 h after inoculation and were immunoprecipitated by anti-FLAG beads. Immunoblotting with an anti-pSer antibody revealed that serine phosphorylation of OSK1-FLAG was significantly enhanced in *Xoc*-inoculated plants compared with that in mock-treated and Δ*xopC2*-inoculated plants (Fig. 4b). Furthermore, inoculation of the *xopC2*-complemented strain activated OSK1 phosphorylation *in planta*, while the *xopC2*^D391A^-transformed strain did not (Fig. 4b). These results indicate that OSK1 phosphorylation is enhanced by XopC2 during *Xoc* infection. Interestingly, immunoprecipitated XopC2-FLAG from rice protoplasts showed a much higher ability to phosphorylate His6-OSK1 than XopC2 purified from *E. coli*, whereas XopC2^D391A/N396A^-FLAG from rice protoplasts completely lost its kinase activity (Fig. 4c).

Because OSK1 was phosphorylated by XopC2, the interaction between XopC2 and OSK1 was further explored. In pulldown assays, the lysates of *E. coli* cells co-expressing His6-OSK1 with GST-XopC2 or GST-XopC2^D391A^ were incubated with GST-beads. His6-OSK1 was simultaneously detected with GST-XopC2 and GST-XopC2^D391A^, but not with GST, on GST-beads (Fig. 4d). Furthermore, co-expression of OSK1-NLuc and CLuc-XopC2 fusion proteins in *N. benthamiana* produced a strong luminescence signal, whereas little signal was detected when OSK1-NLuc was co-expressed with CLuc-AvrBs2. As an additional negative control, the expression of CLuc-XopC2 and rice protein pKWI502-NLuc did not produce any signal (Fig. 4e). These results showed that XopC2 interacts with OSK1 in vivo and in vitro.

**Ser^53^ phosphorylation in OSK1 by XopC2 enhances disease susceptibility in rice.** To investigate whether OSK1

phosphorylation has any effect on the stability of OsJAZ proteins, the phosphomimic mutants OSK1^T32D^-FLAG, OSK1^S53D^-FLAG, OSK1^S92D^-FLAG, and OSK1^T149D^-FLAG were transiently co-expressed with OsJAZ9-HA in rice protoplasts. Only OSK1^S53D^-FLAG caused an obviously reduced OsJAZ9-HA accumulation compared with OSK1 co-expression (Fig. 5a), indicating that the S53D mutation in OSK1 promotes the degradation of JAZ proteins. Therefore, we developed an in vitro JAZ ubiquitination system[40–42] to test whether OSK1 phosphorylation at Ser^53^ affects JAZ ubiquitination (Fig. 5b). First, Cullin1a was incubated with UBA3, AXR1, UBC12, RBX1, Rub1, and DCN1 for Cullin1a rubylation and activation. The Rub1-modified OsCullin1a was then incubated with OsUBA1, UBCH5α, HA-ubiquitin, OsCOI1b, OsJAZ9, and different OSK1 variants in the ubiquitination buffer. Only OSK1^S53D^ out of four phosphosite mutants had an enhanced ability to ubiquitylate OsJAZ9 compared with OSK1 (Fig. 5b). Moreover, we revealed that XopC2-HA, but not XopC2^D391A^-HA, co-expressed in rice protoplasts significantly promoted serine phosphorylation of OSK1^T32A/S92A/T149A^-FLAG. By contrast, OSK1^T32A/S53A/S92A/T149A^-FLAG phosphorylation was not altered by XopC2-HA or XopC2^D391A^-HA (Supplementary Fig. 9b). Next, we developed a polyclonal antibody that specifically detects Ser^53^ phosphorylation in OSK1. The in vitro kinase assays showed that XopC2, but not XopC2^D391A^, strongly phosphorylated OSK1 at Ser^53^ (Fig. 5c). Besides, induced expression of XopC2 caused OSK1 phosphorylation at Ser^53^ in the transgenic lines revealed by immunoblotting (Fig. 5d). The infection of the wild-type *Xoc* and C-Δ*xopC2* strains induced obvious Ser^53^ phosphorylation in rice, whereas the Δ*xopC2* and *xopC2*^D391A^-complemented strains did not (Fig. 5e). We further demonstrated that Ser^53^ phosphorylation of OSK1 in the OsJAZ9-HA-NE-2 transgenic line became to be detectable at 6 h after inoculation with the wild-type *Xoc* and C-Δ*xopC2* strains, while no OSK1 phosphorylation occurred after Δ*xopC2* and *xopC2*^D391A^ inoculation (Supplementary Fig. 9c). These results demonstrated that Ser^53^ in OSK1 is a major phosphorylation site by XopC2.

To test whether Ser^53^ phosphorylation in OSK1 contributes to disease susceptibility in rice, we generated the transgenic rice plants expressing phosphomimic OSK1^S53D^ driven by its native promoter and a maize ubiquitin promoter (Supplementary Fig. 9d). After spray inoculation with Δ*xopC2*, the OSK1^S53D^-expressing transgenic plants exhibited significantly greater bacterial populations than the wild-type and OSK1-expressing transgenic plants (Fig. 5f and Supplementary Fig. 9e). Besides,

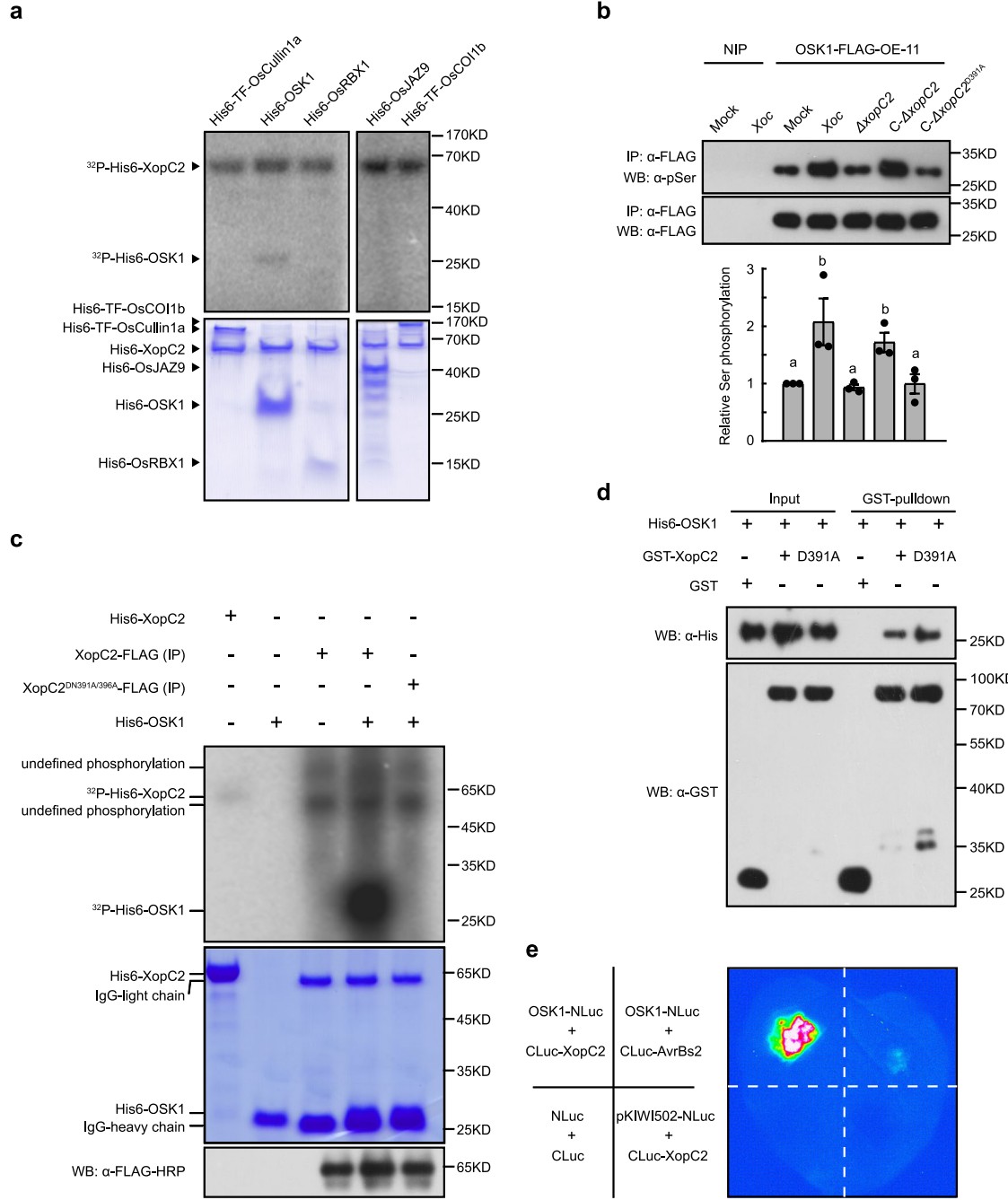

expression of OSK1[S53D], but not of OSK1, significantly compromised stomatal closure in transgenic plants after ΔxopC2 and Xoc infection and enhanced expression of JA-responsive genes after exogenous MeJA treatment (Fig. 5g–i and Supplementary Fig. 9f–h). The results showed that Ser[53] in OSK1 is the preferred and biologically relevant phosphorylation site by XopC2 in vivo. Collectively, our findings indicate that XopC2 phosphorylates OSK1 at Ser[53] to promote bacterial virulence.

### The enhanced binding affinity of phosphomimic OSK1[S53D] to COI1b is dependent on the Arg[13] residue in COI1b.

OSK1 functions as a linker between Cullin1 and F-box proteins. Therefore, we investigated whether the S53D mutation affects the binding affinity of OSK1 to OsCullin1a and OsCOI1b. Co-immunoprecipitation (co-IP) assays showed that OsCOI1b-FLAG immunoprecipitated much more OSK1[S53D]-HA than OSK1-HA while similar amounts of OSK1[S53D]-HA and OSK1-HA were precipitated with OsCullin1a (Fig. 6a and Supplementary Fig. 10a). Besides, microscale thermophoresis (MST) assays showed that GST-OSK1[S53D] had a significantly higher binding affinity to His6-OsCOI1b ($Kd = 12.705 \pm 4.611$ nM) than GST-OSK1 ($Kd = 53.798 \pm 12.263$ nM) (Fig. 6b). The results indicate that the S53D mutation greatly enhances OSK1 binding to OsCOI1b, but does not alter its binding affinity to OsCullin1a.

To understand how Ser[53] phosphorylation enhances the binding affinity of OSK1 to OsCOI1b, structure modeling was performed for OsCOI1b-OSK1-Cullin1-RXB1 and multiple F-box-OSK1 complexes (Supplementary Fig. 10b, c). Unlike F-box proteins OsTIR1 and D3, AtCOI1 and OsCOI1b carry the N-terminal flexible tails (Supplementary Fig. 10c). We speculate that the flexible tail of OsCOI1b containing positively charged

**Fig. 4 XopC2 interacts with and phosphorylates OSK1. a** OSK1 was phosphorylated by XopC2 in vitro. His6-XopC2 was incubated with His6-TF-OsCullin1a, His6-OSK1, His6-OsRBX1, His6-OsJAZ9, and His6-TF-OsCOI1b individually in in vitro kinase assays. Protein phosphorylation was detected by autoradiography. Protein loading was indicated by CBB staining. The experiment was repeated 3 times with similar results. **b** In vivo OSK1 phosphorylation was enhanced during Xoc infection. The OSK1-FLAG-OE-11 transgenic line overexpressing OSK1-FLAG was inoculated with 10 mM $MgCl_2$ (mock), Xoc RS105 (Xoc), $\Delta xopC2$, C-$\Delta xopC2$, and C-$\Delta xopC2^{D391A}$ strains. OSK1-FLAG was immunoprecipitated from total protein extracts of inoculated leaves at 2 dpi. Phosphorylated and total OSK1-FLAG proteins were detected with anti-phosphoserine and anti-FLAG antibodies, respectively. Data are shown as means ± SE ($n = 3$ independent experiments). The chart shows Ser phosphorylation intensities relative to total OSK protein evaluated by Photoshop with three independent repeats. The letters (a, b) indicate a statistically significant difference in OSK1 phosphorylation after inoculation of different strains (one-way ANOVA, Duncan's multiple range test). **c** OSK1 was strongly phosphorylated by XopC2 in a semi-in vitro kinase assay. XopC2-FLAG and its kinase-defective variant XopC2$^{D391A/N396A}$-FLAG were expressed in rice protoplasts and immunoprecipitated with anti-FLAG M2 agarose beads, which were incubated with His6-OSK1 in a kinase assay. Protein phosphorylation was detected by autoradiography. The protein loading was indicated by CBB staining. FLAG-tagged proteins bound to the beads were detected by immunoblotting with an anti-FLAG-HRP antibody. The experiment was repeated 3 times with similar results. **d** In vitro GST-pulldown assay to detect the His6-OSK1 and GST-XopC2 interaction. His6-OSK1 was co-expressed with GST, GST-XopC2, and GST-XopC2$^{D391A}$ in E. coli. GST-XopC2- and GST-precipitated complexes were detected before (Input) and after affinity purification (pulldown) by immunoblotting with anti-His and anti-GST antibodies. The experiment was repeated 3 times with similar results. **e** Luciferase complementation imaging to detect in vivo interaction of OSK1 and XopC2. A strong luminescence signal was recorded in N. benthamiana leaves co-expressing OSK1-NLuc and CLuc-XopC2 at 2 days post agroinfiltration. Little or no luminescence was present in negative controls co-expressing NLuc and CLuc, OSK1-NLuc and CLuc-AvrBs2, and pKIWI502-NLuc and CLuc-XopC2, respectively.

Arg[9], Arg[10], and Arg[13] residues is sterically close to Ser[53] in OSK1, and is involved in the interaction with pSer[53] in OSK1 (Fig. 6c). To test this hypothesis, in vitro GST pulldown assays were performed using purified His6-tagged OsCOI1b$^{R9A}$, OsCOI1b$^{R10A}$, and OsCOI1b$^{R13A}$. Compared with OsCOI1b, OsCOI1b$^{R9A}$, and OsCOI1b$^{R10A}$ that bound to OSK1$^{S53D}$ more tightly than to OSK1, OsCOI1b$^{R13A}$ did not exhibit an enhanced affinity to OSK1$^{S53D}$ (Fig. 6d), indicating that Arg[13] is essential for the enhanced binding of OsCOI1b to OSK1$^{S53D}$. These results showed that Ser[53] phosphorylation enhances the binding affinity of OSK1 to COI1b and promotes the formation of SCF$^{OsCOI1b}$ complex, which depends on Arg[13] in OsCOI1b.

## Discussion

Phytopathogenic bacteria utilize effectors to promote infection by disarming plant immunity and regulating plant cellular processes[2,43]. In this study, we provide evidence that XopC2 is a type of T3S effector kinase. XopC2 specifically phosphorylates a highly conserved adaptor protein OSK1 in SCF complexes. The phosphorylation of OSK1 at Ser[53] enhances its binding affinity to OsCOI1b, which targets OsJAZ for ubiquitination and proteosome-mediated degradation. As a consequence, JA signaling is promoted to suppress stomatal closure, which facilitates Xoc infection.

Multiple canonical and non-canonical bacterial effector kinases have been identified to phosphorylate host target proteins and inhibit host immunity[8,12,44–46]. Despite no apparent similarity to any identified effector kinases at the primary sequence level, XopC2 and its homologs were predicted to possess a putative catalytic motif and P-loop-like ATP-binding motif through PSI-BLAST and HHpred searches (Fig. 1a and Supplementary Fig. 1). The kinase activity of XopC2 was subsequently confirmed by in vitro kinase assays, which demonstrated that XopC2 has the ability to autophosphorylate and phosphorylate OSK1 (Fig. 1c and Fig. 4a, c). Besides the N-terminal unordered region, XopC2 possesses a much longer kinase domain than canonical kinases (Fig. 1b; Supplementary Fig. 1). Sequence alignment showed that the kinase domain of XopC2 greatly differs from those of previously identified bacterial effector kinases except the conserved catalytic triad (Fig. 1), indicating that XopC2 and its homologs represent a novel family of kinase proteins.

Recombinant XopC2 exhibited a much weaker kinase activity than immunoprecipitated XopC2 expressed in rice protoplasts (Fig. 4a, c), suggesting that full activity of XopC2 may require one or more host co-factors. Alternatively, in vitro purified XopC2 may not be correctly folded or properly modified. The requirement of host factors for maximal activity has been documented for other kinases or kinase-fold-like effector proteins. For example, HSP90 is essential for the kinase activity of HopBF1[8]. Similarly, ubiquitin-binding stimulates the kinase activity of OspG[47]. Recent studies showed that calmodulin is a vital co-factor for the glutamylase activity of SidJ, a kinase-fold-like effector protein[48]. The nature of the putative co-factors will be one focus of further study.

Disabling host stomatal immunity is often a prerequisite for successful entry into leaf tissues by various phytopathogenic bacteria including P. syringae[49]. The pathogen suppresses stomatal defense through activating JA signaling with the action of multiple virulence factors, including coronatine, HopZ1, and HopX1[33,34]. X. oryzae pathovars do not produce coronatine[50], nor does it encode homologs of HopZ1 and HopX1. Our data clearly showed that XopC2 in Xanthomonads is functionally equivalent to these effectors in suppressing stomatal immunity.

The SCF ligase complexes, which are conserved functionally and structurally in mammals and in plants[51,52], are usually comprised of a ubiquitin-conjugating enzyme E2, a RING finger protein Rbx1, a scaffold protein Cullin1, a crucial adaptor Skp1, and a variable F-box protein[51]. In mammals, Skp1 links the core Cullin-RING skeleton to diverse F-box proteins and thus forming different SCF E3 ligase complexes to mediate diverse protein degradation[51]. Although 32 Skp1 homologs are encoded in rice[53], OSK1 shares the highest sequence similarity with mammal Skp1 and functions as a major adaptor protein in multiple hormone signaling pathways[54]. Here, we revealed that OSK1 is the only protein in the SCF$^{COI1}$ complex that was specifically phosphorylated by XopC2 (Fig. 4a, c; Supplementary Fig. 8a). The physical interaction of XopC2 and OSK1 was subsequently confirmed by pulldown and luciferase complementation imaging assays (Fig. 4d, e). However, we failed to detect the interaction by co-immunoprecipitation probably because of the transient nature of kinase-substrate interactions, which often prevents the identification of kinase substrates in co-IP assays[55]. The OSK1-coupled SCF complexes regulate major phytohormone signaling pathways by degrading key regulators, such as D53 in strigolactone signaling[56]. Interestingly, expression of the JA-responsive genes, but not auxin-responsive and strigolactone-responsive genes, was enhanced in the XopC2-expressing transgenic seedlings (Fig. 3a, b, and Supplementary Fig. 5h, i), indicating that XopC2-mediated OSK1 phosphorylation specifically promotes JA signaling.

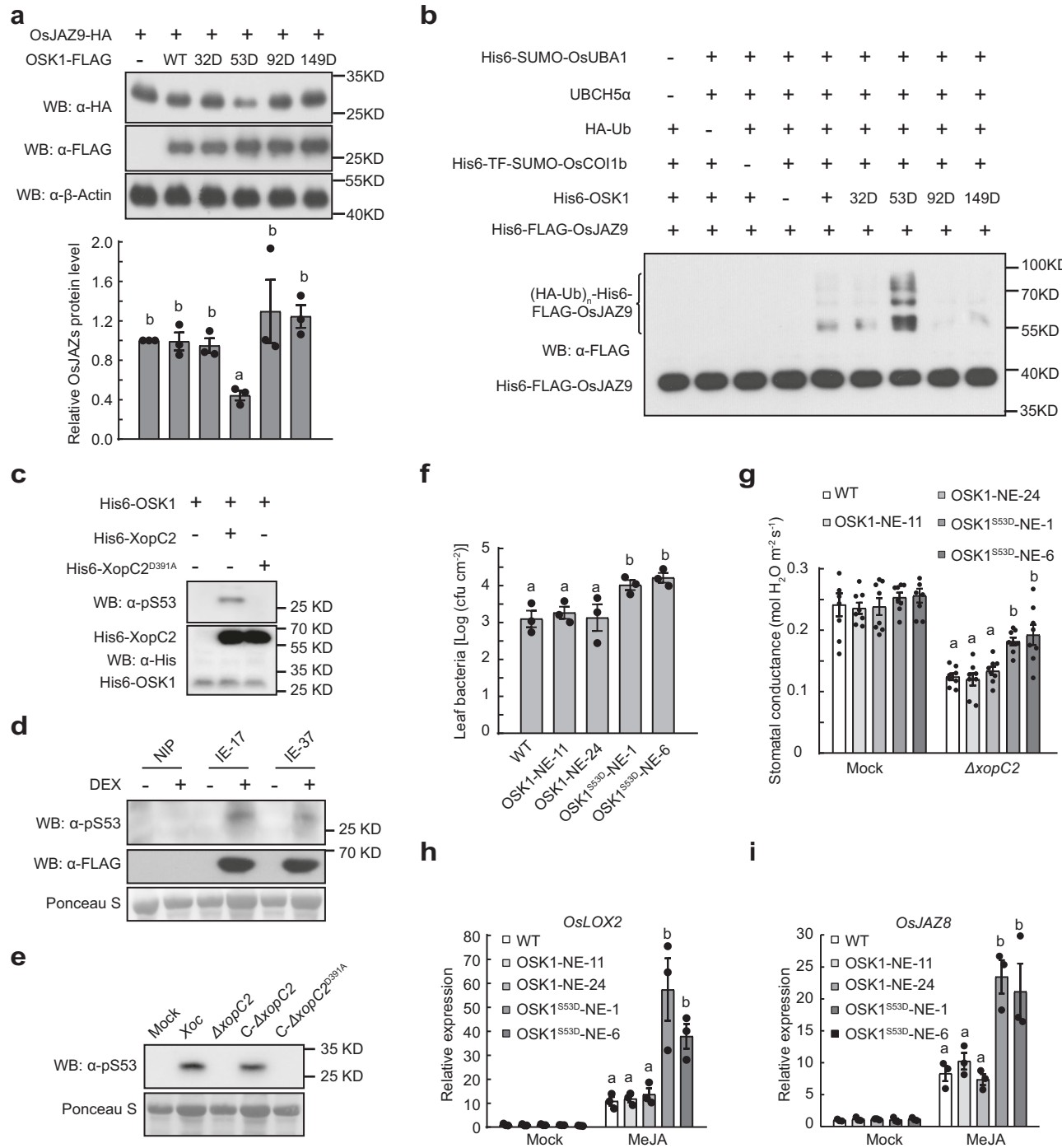

Furthermore, our data showed that much more OSK1 in rice was phosphorylated after *Xoc* infection compared with *ΔxopC2* infection (Fig. 4b), indicating that *Xoc*-secreted XopC2 enhances OSK1 phosphorylation during infection. Subsequently, four phosphosites were identified in OSK1 by in vitro kinase assays (Supplementary Fig. 8d). Among four phosphomimic OSK1 variants, only OSK1$^{S53D}$ promoted SCF$^{OsCOI1b}$-mediated OsJAZ9 ubiquitination and degradation (Fig. 5a, b). Ser$^{53}$ phosphorylation in OSK1 catalyzed by XopC2 was confirmed via in vitro kinase assays and was detected in vivo by immunoblotting with a specific anti-pSer$^{53}$ antibody (Fig. 5c–e). Time course assays showed that XopC2-mediated Ser$^{53}$ phosphorylation in OSK1 occurred at the very early infection stage (Supplementary Fig. 9c), which is consistent with this finding that OsJAZ9

degradation became evident at 6 h post-*Xoc* inoculation (Fig. 3d; Supplementary Fig. 6b). These results indicate that XopC2-mediated Ser$^{53}$ phosphorylation in OSK1 is involved in JAZ degradation and subsequent suppression of stomatal immunity. More convincingly, the transgenic rice lines with OSK1$^{S53D}$ expression exhibited an attenuated stomatal immunity and enhanced JA signaling and disease susceptibility (Fig. 5f–i; Supplementary Fig. 9f, h). We showed that the S53D mutation dramatically enhanced the binding affinity of OSK1 to OsCOI1b, but not to OsCullin1a (Fig. 6a–b; Supplementary Fig. 10a). The Ser$^{53}$ residue is highly conserved in rice Skp1-like homologs and ASK1 in Arabidopsis. Based on structure modeling, we speculate that the flexible N-terminal tail of OsCOI1 with no predicted 3-D structure might be sterically close to the Ser$^{53}$ residue of OSK1

**Fig. 5 XopC2 phosphorylates OSK1 at Ser53 to enhance disease susceptibility in rice. a** OsJAZ9-HA accumulation was significantly reduced when co-expressing with OSK1$^{S53D}$-FLAG compared with expressing alone or with FLAG-tagged OSK1, OSK1$^{T32D}$, OSK1$^{S92D}$, and OSK1$^{T149D}$ in rice protoplasts. The chart shows the OsJAZ9-HA levels normalized to $\beta$-OsActin as evaluated by Photoshop. **b** OSK1$^{S53D}$ promoted SCF$^{OsCOI1b}$-mediated OsJAZ9 ubiquitination. OsCullin1a was first incubated with Rub1, UBA3/AXR1, UBC12, RBX1, Upl1, and OsDCN1. The aliquots of the Rub1-modified OsCullin1a were then incubated with OsUBA1 (E1), UBCH5α (E2), HA-ubiquitin (Ub), OsCOI1b, OSK1, and OsJAZ9 in the ubiquitination buffer. Upl1 was used to remove TF-SUMO tag. OsJAZ9 ubiquitination was detected with an anti-FLAG antibody. **c, d** OSK1 was phosphorylated at Ser53 by XopC2 in vitro and in vivo. His6-XopC2 or His6-XopC2$^{D391A}$ was incubated with His6-OSK1 for in vitro kinase assays (**c**). The IE transgenic lines were treated with DEX (30 μM) or buffer (mock) before protein extraction (**d**). **e** OSK1 was phosphorylated at Ser53 in rice after *Xoc* infection. The wild-type plants were inoculated as described in Fig. 4b. In **c–e**, Ser53 phosphorylation was detected by immunoblotting with an anti-pSer53 polyclonal antibody. **f** Bacterial population sizes in the Δ*xopC2*-inoculated leaves of the wild-type and OSK1-NE-11/24 and OSK1$^{S53D}$-NE-1/6 transgenic seedlings expressing OSK1 and OSK1$^{S53D}$ driven by the native promoter. The wild-type and transgenic rice lines were spray-inoculated as described in Fig. 2a. **g** Stomatal conductance in the leaves of the wild-type, OSK1-NE and OSK1$^{S53D}$-NE transgenic plants after Δ*xopC2* challenging. The experiment was performed as described in Fig. 2b. **h, i** MeJA-induced expression of the JA-responsive gene *OsLOX2* (**h**) and *OsJAZ8* (**i**) in the wild-type, OSK1-NE, and OSK1$^{S53D}$-NE transgenic plants. Data are shown as means ± SE (*n* = 3 independent experiments in **a**; *n* = 3, 8, 3 and 3 technical replicates per measurement in **f**, **g**, **h**, and **i**, respectively). All experiments were independently repeated 3 times with similar results in the panels **a–i**. The letters (a–b) in the panels **a** and **f-i** indicate statistically significant differences as revealed by one-way ANOVA, Tukey's honest significance test.

(Fig. 6c), and therefore, electrostatic interaction occurs between pSer53 in OSK1 and positively charged residues at the N-terminus of OsCOI1b. The hypothesis was confirmed by mutation analysis of OsCOI1b (Fig. 6d), showing that OsCOI1b$^{R13A}$ lost the enhanced binding to OSK1$^{S53D}$. The data indicate that negative charges generated by Ser53 phosphorylation cause OSK1 more attractive to the positively charged Arg13 residue of OsCOI1b.

Based on these findings, a working model is proposed for the virulence mechanism of XopC2 (Fig. 7). Without pathogen challenge, the conserved OsCullin1a-OSK1 skeleton orderly pairs with different F-box proteins to maintain protein homeostasis in rice. During *Xoc* infection, the secreted XopC2 phosphorylates OSK1 at Ser53, and thus, specifically enhances the interaction between OSK1$^{P S53}$ and the F-box protein OsCOI1b. As a consequence, ubiquitination and degradation of JAZ proteins are enhanced, which activates JA signaling and inhibits stomatal immunity. To our knowledge, it is the first report of a novel type of bacterial effector kinase that targets the highly conserved adaptor protein OSK1/ASK1 in the SCF complex. Since the SCF complexes and Skp1 homologs are highly conserved in mammals and in plants, this mechanism is possibly universal in other pathogenic bacteria, consistent with the presence of XopC2 beyond plant pathogens (Supplementary Fig. 1).

Interestingly, OsJAZ9-HA was almost completely degraded in *Xoc*-inoculated rice leaves at 24 h post-inoculation (Fig. 3d), suggesting that XopC2 promotes OsJAZ9-HA degradation not only in guard cells but also in other tissue cells. Therefore, XopC2, besides inhibition of stomatal immunity, might have other virulence function(s) in non-stomatal cells. It has been reported that XopH activates both JA and ethylene hormone signaling and suppresses host immunity in host cells[25]. It will be interesting to investigate whether XopC2 also serves as a virulence factor using other mechanisms.

In summary, we uncover a family of T3S effector kinases in pathogenic bacteria. Although OSK1 is a universal adaptor protein in SCF complexes for multiple hormone signaling pathways, the phosphorylation of the specific residue Ser53 in OSK1 by XopC2 precisely manipulates plant immunity through activating JA signaling rather than interfering with plant growth and development through other hormone actions. Our findings highlight a novel virulence mechanism for bacterial pathogens and pave a road to genetic engineering for disease-resistant plants.

## Methods

**Plant materials and bacterial strains.** *Oryza sativa* ssp. *japonica* cv. Nipponbare was used as the wild type and for generating transgenic plants. *Nicotiana*

*benthamiana* plants were grown at 25 °C under a 12-h light cycle. *Escherichia coli* and *Agrobacterium tumefaciens* strains were cultured in LB medium (1% tryptone, 0.5% yeast extract, 1% NaCl) at 37 °C and 28 °C, respectively. *Xanthomonas oryzae* pvs. *oryzae* and *oryzicola* were cultured in NA medium (0.5% tryptone, 0.1% yeast extract, 0.3% beef extract, and 1% sucrose) at 28 °C. Antibiotics were used at the following concentrations unless specifically noted, 25 μg ml$^{-1}$ rifampicin, 100 μg ml$^{-1}$ timentin, and 50 μg ml$^{-1}$ kanamycin.

**Transgenic constructs and plant transformation.** The open reading frame (ORF) of *xopC2* was amplified using fast *Pfu* DNA polymerase with the primers listed in Supplementary Table 1 and was then sub-cloned into pUC19-35S::FLAG-RBS after digestion with *Csp*45 I and *Spe* I. The *xopC2* ORF and FLAG-coding sequence was cut from the construct by *Xho* I and *Spe* I and re-ligated into the DEX-inducible expression vector pTA7001. For constitutive expression of *xopC2*, ORF was amplified and sub-cloned into pENTR/D TOPO vector (Invitrogen, Carlsbad, CA, USA). The construct was recombined into a binary vector pGWB11[57] using LR clonase II enzyme mix (Invitrogen). *OsJAZ9* was amplified and cloned into pCAMBIA1301-35S::HA after digestion with *Kpn* I and *Xba* I. To construct a native promoter-driven expression vector, the full-length *OsJAZ9* gene (~3 kb) was amplified from rice genome DNA and sub-cloned into pCAMBIA1300 after *Eco*R I and *Kpn* I digestion. The coding sequences of *OSK1* and *OSK1$^{S53D}$* were amplified from pUC19-35S::OSK1-FLAG and pUC19-35S::OSK1$^{S53D}$-FLAG vectors, respectively, and cloned into pC1305-3FLAG after digestion with *Kpn* I and *Hin*d III. To construct the vectors expressing OSK1 and OSK1$^{S53D}$ driven by its native promoter, the maize ubiquitin promoter was removed from pC1305-OSK1-3FLAG and pC1305-OSK1$^{S53D}$-3FLAG by digestion with *Sac* I and *Kpn* I. The *OSK1* promoter fragment (~1.6-kb) was amplified and was then ligated upstream of *OSK1-3FLAG* or *OSK1$^{S53D}$-3FLAG* via in-fusion cloning. After being confirmed by sequencing, these constructs were transformed into *A. tumefaciens* strain EHA105 through the freeze-thaw method. Transgenic rice plants were generated through *Agrobacterium*-mediated transformation[35]. Briefly, the sterilized rice seeds (cv. Nipponbare) were placed on NBi medium (N6 macro elements, B5 microelements, B5 vitamins, 27.8 mg l$^{-1}$ FeSO$_4$·7H$_2$O, 37.3 mg l$^{-1}$ Na$_2$-EDTA, 500 mg l$^{-1}$ proline, 500 mg l$^{-1}$ glutamine, 300 mg l$^{-1}$ casein hydrolysate, 2 mg l$^{-1}$ 2,4-D, 100 mg l$^{-1}$ inositol, 30 g l$^{-1}$ sucrose and 3 g l$^{-1}$ phytagel) plates for 4 weeks at 28 °C for callus induction. The vigorous calli were transferred onto fresh NBi medium for 7 days and were then cultured in NBco medium (NBi supplied with 100 μM acetosyringone, pH 5.5) for 3 days after incubation with Agrobacterium cultures (OD$_{600}$ = 0.1) for 15 min. Subsequently, the calli were washed thoroughly with sterile water and cultured on NBs medium (NBi supplied with 50 mg l$^{-1}$ hygromycin and 400 mg l$^{-1}$ timentin) for 4 weeks. The antibiotic-resistant calli were transferred onto NBr medium (NBi supplied with 0.5 mg l$^{-1}$ α-naphthalene acetic acid, 3 mg l$^{-1}$ 6-benzylaminopurine, 30 mg l$^{-1}$ hygromycin, and 200 mg l$^{-1}$ timentin) for shoot regeneration. The regenerated shoots were transferred into 1/2× MS medium for rooting and were then planted in the greenhouse.

**Construction of complementation strains for the Δ*xopC2* mutant.** An approximately 2.1-kb DNA fragment containing *xopC2-HA* was generated by fusion PCR with pairs of primers *xopC2*-HA-pVSP61-*Eco*RI-F/*xopC2*-HA-pVSP61-mid-R and *xopC2*-HA-pVSP61-mid-F/*xopC2*-HA-pVSP61-*Sal*I-R in Supplementary Table 1, and then cloned into pVSP61[58]. The variant *xopC2$^{D391A}$*-HA was generated via site-directed mutagenesis with the primers *xopC2*-D391A-F and *xopC2*-D391A-R in Supplementary Table 1, and was then sub-cloned into pVSP61. After being confirmed by sequencing, these constructs were introduced into the *Xoc* Δ*xopC2* mutant via tri-parental mating.

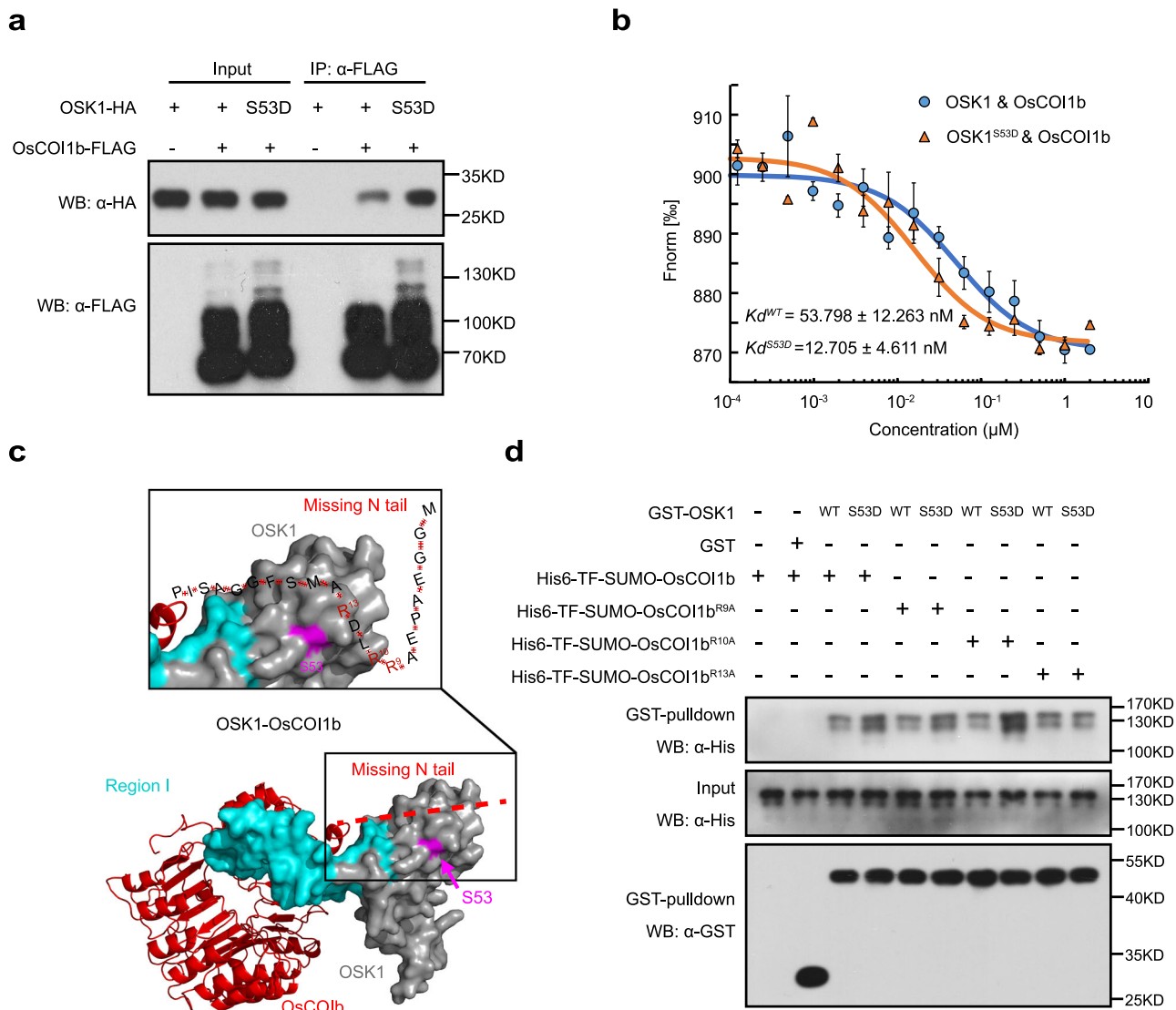

**Fig. 6 The enhanced binding affinity of phosphomimic OSK1$^{S53D}$ to COI1b is dependent on the Arg$^{13}$ residue in COI1b. a** OSK1$^{S53D}$-HA bound to OsCOI1b-FLAG much stronger than OSK1-HA in co-IP assays. OSK1-HA and OSK1$^{S53D}$-HA were individually co-expressed with OsCOI1b-FLAG in rice protoplasts. The immunocomplex and input proteins were analyzed by immunoblotting using anti-HA and anti-FLAG antibodies. The experiments were independently repeated 3 times with similar results. **b** His6-OSK1$^{S53D}$ bound to OsCOI1b much stronger than His6-OSK1 in MST assays. OsCOI1b bound to His6-OSK1$^{S53D}$ with a $Kd$ of 12.705 ± 4.611 nM while to GST-OSK1 with a $Kd$ of 53.798 ± 12.263 nM. Data are shown as means ± SE ($n = 3$ technical replicates). **c** The 3-D structure of OsCOI1b-OSK1 was constructed via homology modeling with the ASK1-COI1-JAZ complex (PDB: 3OGM) as a template. The N-terminal flexible tail of OsCOI1b missing from the template was arbitrarily labeled. The N-terminal tail containing multiple positive-charged Arg residues is predicted to be adjacent to Ser$^{53}$ in OSK1. **d** OsCOI1b$^{R13A}$ had a similar binding affinity to OSK1 and OSK1$^{S53D}$. In vitro-purified His6-TF-SUMO-tagged OsCOI1b$^{R9A}$, OsCOI1b$^{R10A}$ and OsCOI1b$^{R13A}$ were individually incubated with GST-OSK1 and GST-OSK1$^{S53D}$, respectively. After GST pulldown assays, the input and pulldown were detected with anti-His and anti-GST antibodies. The experiments were independently repeated 3 times with similar results.

**SA and JA extraction and quantification**. Endogenous JA and SA were extracted from 4-week-old rice leaves[59]. Briefly, the leaves collected from 4-week-old seedlings (120 mg) were ground to powder in liquid nitrogen. The powder was incubated overnight with 1 ml of extraction buffer (90% methanol, 0.1% formic acid), which was dried by flowing nitrogen gas. The extracted compounds were dissolved in 100 μl of extraction buffer and were then diluted by 5 and 500 times for SA and JA measurement. The SA and JA contents were quantified using plant SA/JA ELISA Kits following the manufacturer's instructions (ZK-8280 for SA and ZK-8014 for JA, Zhen Ke Biological Technology Co., Ltd, Shanghai).

**Chlorophyll content measurement**. The leaves collected from 4-week-old seedlings (100 mg) were ground in liquid nitrogen to powder, which was incubated with 1 ml of acetone for 30 min in the dark. After centrifuge at 13, 400 × $g$ for 10 min, 400 μl of the supernatant was diluted in 3.2 ml acetone. The chlorophyll content was measured by a spectrophotometer and was calculated using the equation, chlorophyll (mg l$^{-1}$) = 20.2 OD$_{645}$ + 8.02 OD$_{663}$[60].

**Virulence assays**. Virulence of $Xoc$ strains to rice was determined by spray and pressure inoculation[61,62]. For pressure inoculation, the wild-type $Xoc$ RS105 and $\Delta xopC2$ strains were re-suspended with 10 mM MgCl$_2$ to an OD$_{600}$ of 0.3. Cell suspensions were pressure-inoculated into rice leaves using needleless syringes. The lesion length, the average of at least ten inoculated leaves, was measured at 14 days after $Xoc$ inoculation.

For spray inoculation, 3-week-old rice seedlings were pretreated with mock or 30 μM DEX supplemented with 0.01% Silwet L77 at 24 h before inoculation. $Xoc$ strains were adjusted to an OD$_{600}$ of 0.8 in 10 mM MgCl$_2$ with 0.01% Silwet L77 and were then sprayed onto rice seedlings. The seedlings were kept in a high humidity chamber (over 90%). After 4 days post-inoculation (dpi), 3 pieces of 5 cm-long leaves with 3 technical repeats were detached from the inoculated seedlings and ground in sterile water after surface sterilization with 75% ethanol. The samples were spread on NA plates after serial dilution. Colony-forming units were counted after 2-day culturing.

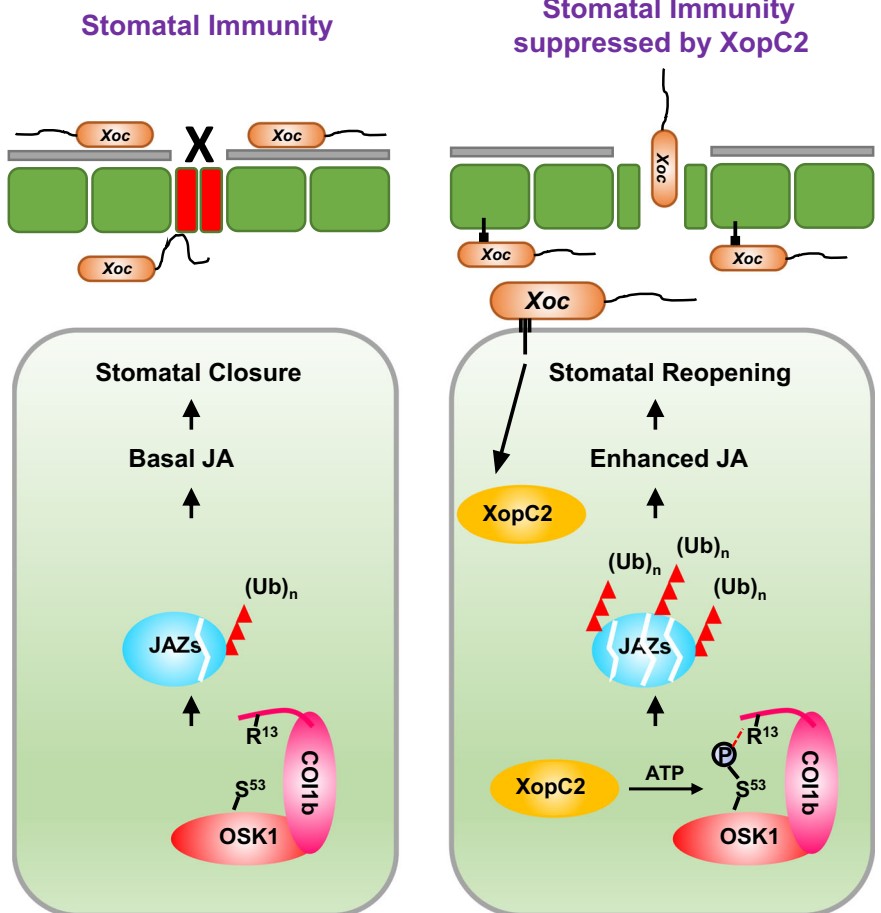

**Fig. 7 A working model for XopC2 function in promoting JA signaling and suppressing stomatal immunity.** During *Xoc* infection, XopC2 is secreted into host cells. As a protein kinase, XopC2 specifically phosphorylates the $Ser^{53}$ residue in OSK1, an essential component of $SCF^{OsCOI1b}$, and thus promotes its binding with OsCOI1b to facilitate the formation of the $SCF^{OsCOI1b}$ complex. This enhances ubiquitination and degradation of JAZ proteins and promotes JA signaling and stomatal reopening. Therefore, stomatal defense is attenuated for the successful entry of the pathogen.

**Stomatal conductance measurement**. Three-week-old *xopC2* IE transgenic seedlings were pretreated with 30 μM DEX or mock at 24 h before spray inoculation of *Xoc* strains. At 2 dpi, stomatal conductance was measured using a photosynthesis system (Model LI-6400XT, Li-Cor Inc., Lincoln, NE, USA)[62]. Briefly, stomatal conductance was measured with a $2 \times 3$ cm leaf chamber supplied with a red-blue LED light source, and parameters were set as 400 μmol mol$^{-1}$ $CO_2$ and 200 μmol m$^{-2}$ s$^{-1}$ photosynthetic photon flux density (PPFD).

**Protein degradation assays in rice protoplasts**. The ORFs of *OsJAZ* genes were amplified with the primers listed in Supplementary Table 1 and sub-cloned into pUC19-*35S::3HA*. The constructed plasmid was confirmed by sequencing. After being isolated using the GoldHi EndoFree Plasmid Maxi Kit (CWBIO), the plasmid was introduced into rice protoplasts prepared from the wild-type and IE-17 transgenic seedlings via PEG-mediated transfection[63]. Briefly, 10 μg of plasmid DNA was incubated with 200 μl protoplasts ($\sim 2.5 \times 10^6$ ml$^{-1}$) in PEG4000/CaCl$_2$ for 10 min. The transfected protoplasts were incubated in the W5 buffer (154 mM NaCl, 125 mM CaCl$_2$, 5 mM KCl and 2 mM MES) supplemented with or without 30 μM DEX and 25 μM MG132 overnight. XopC2-FLAG and OsJAZs-HA were detected by immunoblotting with anti-FLAG (Sigma-Aldrich, F1804, 1:5, 000) and HRP-conjugated anti-HA monoclonal antibodies (Roche, 11667475001, 1:2, 000), respectively. The detection of OsActin using anti-Actin monoclonal antibody (CWBIO, CW0264, 1: 5, 000) was used as a loading control.

The ORF of *OSK1* was amplified with OSK1-FLAG-*Kpn* I-F and OSK1-FLAG-*Xba* I-R (Supplementary Table 1) and was then sub-cloned into pUC19 with the 35S promoter after digestion by *Kpn* I and *Xba* I. The *OSK1* mutant vectors were generated based on pUC19-*35S::OSK1-FLAG* via site-directed mutagenesis[64]. All constructs were confirmed by sequencing. The pUC19-*35S::OsJAZ9-3HA* plasmid was co-transfected with pUC19-*35S::OSK1-FLAG*, its phosphomimic variants, and pUC19 individually. The transfected cells were incubated in the W5 buffer overnight.

**OsJAZ9-HA degradation assay in transgenic rice plants during *Xoc* infection**. Three-week-old transgenic rice seedlings expressing OsJAZ9-HA were challenged with *Xoc* strains by spray inoculation. Three 2 cm-length leaf pieces were collected at indicated timepoints after inoculation and grounded into powder in liquid nitrogen. Total proteins were extracted with 400 μl of SDS sample loading buffer. OsJAZ9-HA was detected by immunoblotting with HRP-conjugated anti-HA monoclonal antibodies (Roche, 11667475001, 1:2, 000).

**Protein expression in *E. coli* and purification**. The ORFs of *xopC2*, *OSK1*, *OsRBX1*, *OsUBA3*, *OsAXR1*, *OsUBC12*, *OsRXB1*, *OsRub1*, and *OsDCN1* were amplified with the primers (Supplementary Table 1) and were then sub-cloned into pET28a. *OsJAZ9* and *OsJAZ9-FLAG* were cloned into pET32b. *OSK1*, *Upl1*, and *xopC2* were cloned into pGEX-4T-3. *OsUBA1* was cloned into pCold-SUMO. *OsCOI1b* and *OsCullin1a* were cloned into pCold-TF. *SUMO-OsCullin1a* and *SUMO-OsCOI1b* fragments were generated using fusion PCR and sub-cloned into pCold-TF carrying His6 tag to generate pCold-TF-*SUMO-OsCullin1a* and pCold-TF-*SUMO-OsCOI1b*. The vectors carrying *xopC2*, *OSK1* and *OsCOI1b* mutants were generated via site-directed mutagenesis. All constructs confirmed by sequencing were transformed into *E. coli* BL21(DE3) for protein expression.

For in vitro protein purification, 100 μM of IPTG was added into cell cultures to induce protein expression at a cell density of $OD_{600} = 0.6$. The cells were further cultured at 16 °C with shaking at 150 rpm overnight. Cell cultures were collected by centrifuge at $1,500 \times g$ for 10 m and were then broken in the lysis buffer containing 50 mM Tris-Cl, pH 8.0, 150 mM NaCl, and 10 mM imidazole for His-tagged proteins or in the binding buffer containing 50 mM Tris-Cl, pH 8.0, and 150 mM NaCl for GST-tagged proteins. Cell debris was removed by centrifuge at $13,400 \times g$ for 10 m at 4 °C, and the supernatants were loaded onto Ni-NTA His·Bind® resin and GST·Bind$^{TM}$ resin (Navogen, EMD Millipore, Billerica, MA), respectively. For His-tagged proteins, the resin was washed twice with lysis buffer and twice with washing buffer (50 mM Tris-Cl, pH 8.0, 150 mM NaCl, and 20 mM imidazole), and was then eluted with 2 ml of elution buffer (50 mM Tris-Cl, pH 7.5, 150 mM NaCl, and 250 mM imidazole). For GST-tagged proteins, the resin was washed four times

with binding buffer and was then eluted with 2 ml of elution buffer (50 mM Tris-Cl, pH 7.5, 150 mM NaCl, and 10 mM glutathione).

**In vitro kinase assay.** In vitro kinase assays were performed as previously described with some modifications[65]. His6-XopC2 and variant proteins (5 μg) were incubated at 28 °C for 1 h in the reaction buffer containing 50 mM Tris-Cl, pH 7.5, 10 mM $MgCl_2$, 1 mM DTT, and [γ-$^{32}$P]ATP. Alternatively, His6-XopC2 and variant proteins (1 μg) were incubated in the reaction buffer with or without 5 μg of substrates at 28 °C for 2 h. Reactions were stopped with 1× SDS sample buffer followed by heating at 95 °C for 5 m. Proteins were separated on 12% SDS-PAGE, and phosphorylation signals were detected via autoradiography using a Storage Phosphor Screen with Typhoon Trio Variable Mode Imager (GE Healthcare, Piscataway, NJ).

**In vivo OSK1 phosphorylation assay.** Six-week-old transgenic rice plants expressing OSK1-FLAG were inoculated with *Xoc* strains. Six leaf discs (5 mm, diameter) were collected from the inoculated leaves at 2 dpi and ground into powder in liquid nitrogen. Total proteins were isolated with the extraction buffer (50 mM Tris-Cl, pH 7.4, 150 mM NaCl, 1 mM EDTA, 1% Triton X-100, protease inhibitor cocktail (CWBIO), 5 mM NaF, and 5 mM $Na_3VO_4$). OSK1-FLAG was immunoprecipitated with anti-FLAG M2 affinity gel from total protein extracts and washed 5 times with 1× PBS buffer supplemented with 2 mM NaF and 0.1 mM $Na_3VO_4$. Precipitated OSK1-FLAG and its phosphorylation were detected by western blotting with an HRP-conjugated anti-FLAG monoclonal (Sigma-Aldrich, A8592, 1:5, 000) and anti-phosphoserine polyclonal antibodies (Millipore, AB1603, 1:5, 000), respectively.

**Determination of OSK1 phosphorylation at Ser$^{53}$ by XopC2.** The *xopC2* ORF and its variant *xopC2$^{D391A}$* were amplified from the pET28a constructs with *xopC2*-Kpn I-F and *xopC2*-Xba I-R and were then sub-cloned into pUC19-35S::3HA. The pUC19-35S::OSK1$^{S32A/S92A/T149A}$-3FLAG and pUC19-35S::OSK1$^{S32A/S53A/S92A/T149A}$-3FLAG plasmids were individually co-transfected into rice protoplasts (~$2.5 \times 10^6$ ml$^{-1}$) with pUC19, pUC19-35S::xopC2-3HA or pUC19-35S::xopC2$^{D391A}$-3HA plasmids. After overnight incubation, total proteins were isolated from the transfected cells with 1 ml of extraction buffer. OSK1$^{S32A/S92A/T149A}$-FLAG and OSK1$^{S32A/53A/S92A/T149A}$-FLAG were immunoprecipitated with anti-FLAG M2 agarose gel and washed 5 times with 1× PBS buffer supplemented with 2 mM NaF and 0.1 mM $Na_3VO_4$. OSK1 phosphorylation was detected by immunoblotting with an anti-phosphoserine polyclonal antibody (Millipore, AB1603, 1:5, 000), while OSK1-FLAG, XopC2-HA, and XopC2$^{D391A}$-HA were detected via immunoblotting with HRP-conjugated monoclonal antibodies. Ser$^{53}$ phosphorylation in OSK1 was specifically detected with an anti-OSK1$^{PS53}$ polyclonal antibody generated by immunizing rabbits with a phospho-peptide, Ac-C-LPNVN(pS)KILSK-NH$_2$ (Abmart, Shanghai, China).

**Semi-in vitro ubiquitination assay.** *E. coli* cell cultures (100 ml) expressing His6-OsJAZ9 were collected and lysed in lysis buffer (50 mM Tris-Cl, pH 8.0, 150 mM NaCl, and 10 mM imidazole). His6-OsJAZ9 in the supernatant was bound to 500 μl of Ni-NTA agarose beads without elution at 4 °C. The beads were then homogeneously re-suspended in 500 μl of suspension buffer (50 mM Tris-Cl, pH 7.4, and 150 mM NaCl) and distributed into centrifuge tubes with 20 μl beads per tube. Total rice protein extracts (TRPE) were isolated from 7-day-old seedlings with the buffer containing 50 mM Tris-Cl, pH 7.4, 150 mM NaCl, 1 mM EDTA, 1% Triton X-100, protease inhibitor cocktail (CWBIO, CW2200, 1: 100), and 50 μM MG132. The ubiquitination assay was performed at 28 °C with 20 μl of His6-OsJAZ9-bound Ni-NTA agarose, 50 μg TRPE$^{-/+}$, 5 μg HA-ubiquitin$^{-/+}$, and 5 μg GST-XopC2$^{-/+}$ as indicated in the reaction buffer containing 50 mM Tris-Cl, pH 7.4, 5 mM ATP, 5 mM $MgCl_2$, protease inhibitor cocktail, 100 μM MG132, and 50 mM coronatine. After incubation for the indicated time, the samples were collected by centrifuge at $100 \times g$ at 4 °C for 1 m. The beads were washed 5 times with 1× PBS buffer (pH 7.4) and were then subjected to heating at 95 °C for 10 m after adding 100 μl of SDS sample loading buffer. His6-OsJAZ9 ubiquitination and GST-XopC2 were detected by immunoblotting with an HRP-conjugated anti-HA antibody (Roche, 11667475001, 1:2, 000) and a GST monoclonal antibody (CWBIO, CW0084, 1:5, 000), respectively. The His6-OsJAZ9 protein levels were detected by CBB staining.

**A refined semi-in vitro ubiquitination assay via immunoprecipitated SCF$^{OsCOI1b}$.** The ORFs of *OsCOI1b* and *OSK1* were cloned into pUC19-35S::FLAG-RBS and pUC19-35S::3HA, respectively, and the ORFs of *OsCullin1a* and *OsRBX1* were cloned into pHBT-35S::HA after amplification with the primers in Supplementary Table 1. After being confirmed by sequencing, the four constructed plasmids were co-transfected into rice protoplasts. About 10 ml of rice protoplasts (~$2.5 \times 10^6$ ml$^{-1}$) were incubated with 500 μg of plasmid DNA (125 μg each construct) for 14 h at 28 °C. Total proteins were extracted with the buffer containing 50 mM Tris-Cl, pH 7.4, 5 mM ATP, 5 mM $MgCl_2$, protease inhibitor cocktail, and 100 μM MG132. SCF$^{OsCOI1b}$ complex was immunoprecipitated by 100 μl of anti-FLAG M2 agarose gel and washed 4 times with 1× PBS buffer. The ubiquitination was performed at 28 °C for

90 m in 40 μl reaction volume containing 10 μl of SCF$^{OsCOI1b}$-bound beads, 100 ng E1$^{-/+}$, 300 ng E2, 2 μg HA-Ub$^{-/+}$, 1 μg His6-OsJAZ9$^{-/+}$, 1 μg GST-XopC2$^{-/+}$ or GST-XopC2 mutant$^{-/+}$ as indicated in the buffer (50 mM Tris-Cl, pH 7.5, 2 mM ATP, 5 mM $MgCl_2$, 0.5 mM DTT, protease inhibitor cocktail, 100 μM MG132, and 50 mM COR). After incubation, the ubiquitination was detected by immunoblotting with HRP-conjugated anti-HA (Roche, 11667475001, 1:2, 000) and HRP-conjugated anti-His antibodies (CWBIO, CW0285, 1:5, 000).

**In vitro JAZ9 ubiquitination assay.** SCF$^{OsCOI1b}$-mediated JAZ9 ubiquitination was performed in vitro[66,67]. His6-TF-SUMO-OsCullin1a (1 μM) was incubated with 500 nM His6-OsUBA3, 500 nM His6-OsAXR1, 1 μM His6-OsUBC12, 1 μM His6-OsRBX1, 1 μM His-OsDCN1, 5 μM His6-OsRub1, and 1 μM GST-Upl1 at 28 °C for 1 h in a 400 μl reaction solution containing 50 mM Tris-Cl, pH 7.5, 2 mM ATP, 5 mM $MgCl_2$, 50 μM coronatine, and 0.5 mM DTT. The reaction mixture was equally divided into centrifuge tubes and was further incubated with different combinations of 50 nM His6-SUMO-OsUBA1, 100 nM UBCH5α, 2 μM HA-ubiquitin, 1 μM His-OsJAZ9-FLAG, 1 μM His6-TF-SUMO-OsCOI1b, and 1 μM His6-OSK1 or His6-OSK1 variants for 90 m. His6-OsJAZ9-FLAG ubiquitination was detected by immunoblotting with an anti-FLAG monoclonal antibody (Sigma-Aldrich, F1804, 1:5, 000).

**Identification of phosphosites in OSK1 via LC-MS/MS.** The phosphosites in OSK1 were determined as described previously[65]. Briefly, in vitro purified His6-OSK1 (50 μg) and His6-XopC2 (25 μg) were incubated together in the buffer containing 50 mM Tris-Cl, pH 7.5, 5 mM ATP, 5 mM $MgCl_2$, and 1 mM DTT for 4 h at 28 °C. Proteins were digested with trypsin at 37 °C for 4 h and separated by a Waters nanoAcquity nano HPLC (Waters, Milford, MA). Nanospray ESI-MS was performed with a Thermo Q-Exactive high-resolution mass spectrometer (Thermo Scientific, Waltham, MA). The phosphorylated sites were analyzed by MS/MS spectra data with Mascot Distiller (Matrix Science; version 2.4).

**Luciferase complementation imaging assay.** Luciferase complementation imaging assay was performed through *Agrobacterium*-mediated transient expression in *N. benthamiana*[68]. Briefly, the ORFs of *xopC2* and *OSK1* were amplified with the primers listed in Supplementary Table 1 and were then cloned into pCAMBIA1300-35S::NLuc and pCAMBIA1300-35S::CLuc, respectively. After being confirmed by sequencing, the plasmids were transformed into *A. tumefaciens* EHA105 cells, and were co-infiltrated into *N. benthamiana* leaves. The LUC images were captured on the infiltrated leaves at 2 days post agroinfiltration immediately after spraying luciferin using a low-light cooled CCD imaging apparatus (Berthold LB985).

**In vitro GST pulldown assay.** GST pulldown assays were conducted following the procedures with minor modifications[69]. Briefly, 40 μl of GST·Bind$^{TM}$ resin was washed by PBS buffer twice before incubating with 10 μg of GST-tagged and His6-tagged proteins. After incubation at 4 °C for 4 h in lysis buffer (50 mM Tris-Cl, pH 7.4, 150 mM NaCl, 1 mM EDTA, and 1% Triton X-100), the supernatant was removed and the beads were thoroughly washed with ice-cold PBS buffer. The proteins bound to the beads were released with 100 μl of SDS sample loading buffer and were then subject to boiling for 5 min before immunoblotting.

For binding assays between His6-OSK1 and GST-XopC2/GST-XopC2$^{D391A}$, the pGEX-4T-3-xopC2 and pET28a-OSK1 plasmids were co-transformed into *E. coli* BL21 (DE3). At 4 h after induced expression by 100 μM IPTG at OD$_{600}$ = 0.6, cell cultures (50 ml) were collected and broken by the supersonic method in the lysis buffer (50 mM Tris-Cl, pH 8.0, and 150 mM NaCl). The cell lysate was incubated with 40 μl of GST·Bind$^{TM}$ resin for 4 h, and pulldown assays were then performed as described above.

**In vivo co-immunoprecipitation assay.** Co-immunoprecipitation was performed using anti-FLAG M2 affinity gel (Sigma-Aldrich) following the manufacturer's instructions. Briefly, anti-FLAG M2 affinity gel suspension (40 μl) was transferred to centrifuge tubes and was then washed twice with PBS buffer. Total cell lysate (1 ml) was added to the resin and then incubated at 4 °C for 4 h. The resin was thoroughly washed 5 times with 1 ml of ice-cold PBS buffer. The SDS sample loading buffer was then added to beads and heated to 100 °C for 5 min before immunoblotting. Total proteins and the immuno-precipitated proteins were detected using anti-FLAG (Sigma-Aldrich, F1804, 1:5, 000) and anti-HA-HRP (Roche, 11667475001, 1:2, 000) antibodies.

**Microscale thermophoresis assay.** Microscale thermophoresis (MST) assay was performed using a NanoTemper Monolith$^{TM}$ NT.115 instrument (Munich, Germany). His6-OSK1, His6-OSK1$^{S53D}$, and His6-TF-SUMO-OsCOI1b were further purified with a prepacked Superdex$^{TM}$ 200 10/300 GL Tricorn$^{TM}$ high-performance SEC column using AKTA chromatography system (GE Healthcare). The proteins were dialyzed with 1× PBS buffer three times at 4 °C. His6-TF-SUMO-OsCOI1b (2 μM, 100 μl) was incubated with 6 μl of the fluorescence dye RED-NHS 2nd generation for 30 m in 25 °C. Labeled His6-TF-SUMO-OsCOI1b (10 μl) was individually mixed with an equal volume of serially-diluted His6-OSK1 or

His6-OSK1$^{S53D}$ in 1× PBS buffer supplied with 0.05% Tween 20. After incubation at 25 °C for 10 m, the mixtures were loaded onto standard-treated silica capillaries (NanoTemper) and fluorescence was measured on 20% LED power and 20% IR-laser power.

**qRT-PCR analysis.** Six-day-old wild-type and *xopC2* transgenic rice seedlings were sprayed with 30 μM DEX for 24 h followed by treatment with 50 μM MeJA or mock control for 6 h. The seedlings were collected for RNA isolation using an ultrapure RNA extraction kit according to the manufacturer's protocol (CWBio). Complementary DNA (cDNA) was synthesized by a reverse transcription system (Takara, Dalian, China) using total RNA as a template. Quantitative RT-PCR (qRT-PCR) was performed using an ABI PRISM® 7500 Sequence Detection System (Applied Biosystems, Carlsbad, CA, USA). The gene expression levels were calculated based on three repeats and were normalized against the expression of *OsActin1* (Os03g0718100). The primers used for qRT-PCR were listed in Supplementary Table 1.

**Statistics and reproducibility.** The data in each figure are from representative experiments that were independently repeated at least three times. Statistical analyses were performed with a two-sided *t*-test, one-way ANOVA, following multiple comparisons of means with Tukey's honest significance test or Duncan's multiple range test.

**Reporting summary.** Further information on research design is available in the Nature Research Reporting Summary linked to this article.

## Data availability

All data and materials generated in the article are available from the corresponding author as request. Source data for figures and supplementary figures are provided in a Source data file. Source data are provided with this paper.

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

## Acknowledgements

We thank Junfeng Liu at China Agricultural University for valuable comments. Jianmin Zhou at Institute of Genetics and Development Biology, Chinese Academy of Sciences, and Zichao Li at China Agricultural University for the pUC19-35S-3FLAG-RBS, pCMABIA1300-NLuc/CLuc, and pC1305-3FLAG plasmids. The work is supported by the National Natural Science Foundation of China grants 31770140, 31801695, and 31630064 to W.S. and S.W., and the Programme of Introducing Talents of Discipline to Universities B13006 to W.S.

## Author contributions

S.W. and W.S. conceived the study, and S.W., S.L., W.S., Z.Q.L., and S.Y.H. contributed to experimental design. S.W., S.L., Q.L., J.W., X.X., S.Z., Y.W., D.L., and J.X. performed the research and analyzed data. S.W., W.S., Z.Q.L., and S.Y.H. wrote and edited the article.

## Competing interests

The authors declare no competing interests.
