## [Peer Review File · Nature Communications]

A bacterial kinase phosphorylates OSK1 to suppress stomatal immunity in riceREVIEWER COMMENTS

Reviewer #1 (Remarks to the Author):

The manuscript describes that *Xanthomonas* effector XopC2 belongs to a new family of protein kinases, and induces stomatal reopening by promoting the JA signaling through phosphorylation of OSK1. The authors found that XopC2 contains putative P-loop motif and catalytic motif of protein kinases. In fact, XopC2 possesses the autophosphorylation activity. Rice transgenic plants expressing XopC2 showed enhanced susceptibility when they were inoculated by spray but not pressure infiltration, indicating that XopC2 functions at the initial step of natural infection of Xoc. Expression of XopC2 in rice suppresses stomatal closure triggered by Xoc infection. MeJA-induced expression of JA-responsive marker genes was enhanced in the XopC2-OX plants. These results suggest that XopC2 promotes JA signaling.

In addition, XopC2 promotes JAZ protein degradation through ubiquitination of the JAZ proteins by the SCF-OsCOI1 complex, which depends on the kinase activity of XopC2. XopC2 interacts with and phosphorylates Ser53 of OSK1, a member of the SCF-OsCOI1 complex. Expression of the phosphomimic mutant S53D of OSK1 enhanced ubiquitination and degradation of OsJAZ9. Furthermore, rice plants overexpressing S53D mutant of OSK1 exhibited reduction of disease resistance and stomatal closure. These indicate that XopC2 phosphorylates OSK1 at Ser53 to promote bacterial virulence. Finally, the S53D mutation of OSK1 enhanced the interaction with OsCOI1b, which may promote the formation of SCF-OsCOI1b complex. Thus, the manuscript provides a novel regulatory mechanism for effector-mediated suppression of host immunity. I think that the data shown here have high impact for the broad range of readers. I have only few comments as below.

Major comments

1) From the data of Fig 4b and Fig 5e, OSK1 was phosphorylated at 2 days after infection with Xoc. However, Fig3d indicated that OsJAZ9 was almost completely degraded at 24h after infection. I think that OSK1 should be phosphorylated before degradation of OsJAZ9. Did the authors detect the phosphorylation of OSK1 before the degradation of OsJAZ9?

2) The data presented here show that XopC2 phosphorylates OSK1 to induce stomatal reopening. However, the authors used the 35S promoter to generate transgenic plants expressing OSK1 or OsJAZ9. Thus, OSK1 and OsJAZ9 were likely expressed in all cells of these transgenic plants. Because OsJAZ9 was almost completely degraded after infection with Xoc, the JA signaling could be activated not only at guard cell but also at other cells. Therefore, I think a possibility that XopC2 also could suppress host immunity in stomata-independent manner. Please discuss the possibility.

Reviewer #2 (Remarks to the Author):

In this manuscript, Wang et al. report XopC2 as an atypical protein kinase which can activate plant JA signaling and perturb stomatal defense. I think the topic of this study is important, and the findings add substantial knowledge toward a mechanistic understanding of the plant-pathogen interactions. Overall, the manuscript is clearly written and the experiments are well designed and executed. However, there are a few issues need the authors to address. For example, Result section: Fig1c shows much less reduction in autophosphorylation activity caused by N396A mutation than by D391A, what is the cause/biological relevance of this difference? Lines 163-171 and Figs4b, regarding the transgenic plants, I wonder whether ectopic expression of XopC2 brought about any toxic effects on plant growth and development? To my eyes the leaves of IE17 are narrower than those of WT. How about those of OE-1 or of OE-10 plants? Whether expressing XopC2 affected stomatal development? It seems to me that XopC2 expression affects the leaf color, so it will be informative if the authors can test the chlorophyll contents as well as carry out JA-induced leaf senescence assays given XopC2 could activate JA pathway. Also, I hope the authors can construct and observe the transgenic plants expressing XopC2 N396A, XopC2 D391A, respectively, along with those expressing XopC2. For example, whether the mutations could diminish bacterial disease severity? Fig2b the DEX-induced XopC2 expression compromises stomatal closure, whether this function of XopC2 is dependent to intact D391/N396? It will be helpful if the author can examine the gs at least at three time points to observe the possible effects of XopC2 on stomatal closure and or stomatal reopening. Whether XopC2 can repress the expression of genes involved in SA biosynthesis and the accumulation of SA through JA pathway? Whether JA/SA content changes in response (are important) to XopC2?

Introduction section: the authors should introduce stomatal immunity properly to better establish the background of their study.

Abstract (line 50): I am not convinced that XopC2 “precisely” activates JA signaling.

REVIEWER COMMENTS

Reviewer #1 (Remarks to the Author):

The manuscript describes that *Xanthomonas* effector XopC2 belongs to a new family of protein kinases, and induces stomatal reopening by promoting the JA signaling through phosphorylation of OSK1. The authors found that XopC2 contains putative P-loop motif and catalytic motif of protein kinases. In fact, XopC2 possesses the autophosphorylation activity. Rice transgenic plants expressing XopC2 showed enhanced susceptibility when they were inoculated by spray but not pressure infiltration, indicating that XopC2 functions at the initial step of natural infection of *Xoc*. Expression of XopC2 in rice suppresses stomatal closure triggered by *Xoc* infection. MeJA-induced expression of JA-responsive marker genes was enhanced in the XopC2-OX plants. These results suggest that XopC2 promotes JA signaling.

In addition, XopC2 promotes JAZ protein degradation through ubiquitination of the JAZ proteins by the SCF-OsCOI1 complex, which depends on the kinase activity of XopC2. XopC2 interacts with and phosphorylates Ser53 of OSK1, a member of the SCF-OsCOI1 complex. Expression of the phosphomimic mutant S53D of OSK1 enhanced ubiquitination and degradation of OsJAZ9. Furthermore, rice plants overexpressing S53D mutant of OSK1 exhibited reduction of disease resistance and stomatal closure. These indicate that XopC2 phosphorylates OSK1 at Ser53 to promote bacterial virulence. Finally, the S53D mutation of OSK1 enhanced the interaction with OsCOI1b, which may promote the formation of SCF-OsCOI1b complex. Thus, the manuscript provides a novel regulatory mechanism for effector-mediated suppression of host immunity. I think that the data shown here have high impact for the broad range of readers. I have only few comments as below.

Major comments

1) From the data of Fig 4b and Fig 5e, OSK1 was phosphorylated at 2 days after infection with *Xoc*. However, Fig3d indicated that OsJAZ9 was almost completely degraded at 24h after infection. I think that OSK1 should be phosphorylated before

degradation of OsJAZ9. Did the authors detect the phosphorylation of OSK1 before the degradation of OsJAZ9?

AUTHOR RESPONSE: Thank you for your great suggestions! In this revised manuscript, we detected the phosphorylation of OSK1 at Ser⁵³ and the protein level of OsJAZ9-HA at 6, 12 and 24 hours post bacterial inoculation (hpi) in the transgenic rice plants expressing OsJAZ9-HA driven by its native promoter (Fig. 3d and Supplementary Fig. 9c). We demonstrated that the phosphorylation of OSK1 at Ser⁵³ was detectable at 6 h after inoculation of the wild-type and complemented *Xoc* strains, and at the same time the degradation of OsJAZ9 was also detected. The phosphorylation of OSK1 at Ser⁵³ was gradually increased and the protein level of OsJAZ9-HA was gradually decreased thereafter. The results indicate that the phosphorylation of OSK1 at Ser⁵³ occurs at very early infection stage and induces OsJAZ9 degradation.

2) The data presented here show that XopC2 phosphorylates OSK1 to induce stomatal reopening. However, the authors used the 35S promoter to generate transgenic plants expressing OSK1 or OsJAZ9. Thus, OSK1 and OsJAZ9 were likely expressed in all cells of these transgenic plants. Because OsJAZ9 was almost completely degraded after infection with *Xoc*, the JA signaling could be activated not only at guard cell but also at other cells. Therefore, I think a possibility that XopC2 also could suppress host immunity in stomata-independent manner. Please discuss the possibility.

AUTHOR RESPONSE: A great concern! To avoid the side-effect brought by OSK1 and OsJAZ9 overexpression, we generated the transgenic rice plants expressing OSK1 and OsJAZ9 under the native promoters. Using these transgenic plants, we repeated all related experiments and obtained the similar results to those from the overexpressing transgenic plants. In Supplementary Fig. 7b (Fig. 3d in the former version), we showed a western blot image with long-time exposure. The image showed that OsJAZ9-HA was largely degraded, although detectable, in the OsJAZ9-HA-overexpressing transgenic plants at 24 hours after inoculation with the wild-type and complemented *C-ΔxopxC2* strains. Consistently, the transgenic rice plants with OsJAZ9-HA expression under the native promoter exhibited a similar OsJAZ9

degradation rate as the OsJAZ9-HA-overexpressing transgenic plants (Fig. 3d), indicating that XopC2 promotes the degradation of OsJAZ9-HA not only in the guard cells but also in other cells.

Based on these results, we totally agree with the reviewer on the speculation that XopC2 also suppresses host immunity in stomata-independent manner. A previous study showed that XopH activates both JA- and ET-mediated signaling to suppress host immunity (Blüher et al., 2017). We hypothesize that XopC2 might have other pathogenic function(s) in the non-stomatal cells rather than suppressing stomatal closure. It will be an interesting topic to figure out the unidentified virulence functions of XopC2 in the future. We discussed this speculation and possibility in the revised manuscript.

Reference:

Blüher, D. *et al.*, A 1-phytase type III effector interferes with plant hormone signaling. *Nat Commun* **8**, 2159 (2017).

Reviewer #2 (Remarks to the Author):

In this manuscript, Wang et al. report XopC2 as an atypical protein kinase which can activate plant JA signaling and perturb stomatal defense. I think the topic of this study is important, and the findings add substantial knowledge toward a mechanistic understanding of the plant-pathogen interactions. Overall, the manuscript is clearly written and the experiments are well designed and executed. However, there are a few issues need the authors to address. For example, Result section: Fig1c shows much less reduction in autophosphorylation activity caused by N396A mutation than by D391A, what is the cause/biological relevance of this difference?

AUTHOR RESPONSE: Thank you for your comments! The majority of protein kinases have a conserved catalytic motif, which is responsible for transferring γ -phosphate group of ATP to different substrates. According to sequence alignment published in the previous study (Figure 4 in Kannan et al., 2007; the figure is attached as follows), the Asp residue in the catalytic motif is absolutely conserved in all identified kinase families; while the Asn residue is replaced by other amino acid residues in a few active

kinase families, such as MTRK family (N to S), MalK family (N to Q) and Alpha family (N to G/V). Based on the crystal structure of ATP-Mg²⁺-NleH (PDB: 4LRJ) (Grishin et al., 2014), the conserved Asp residue (Asp³⁹¹ in XopC2) directly forms hydrogen bonds with ATP and Mg²⁺ and mediates the γ -phosphate transfer. The Asn residue (Asn³⁹⁶ in XopC2) does not directly mediate the transfer of γ -phosphate, but cooperates with a few other residues to keep the second Mg²⁺ ion in the right spatial position. The published results and analyses suggest that the Asp mutation will cause the kinase to lose the catalytic activity, while the Asn mutation may attenuate but not disable the kinase activity. Our results in Fig. 1C were in consistent with these speculations.

Sequence logos depicting conservation of core motifs and neighboring sequences across most kinase families (Cited from Figure 4 in Kannan et al., 2007, PLoS Biol 5, e17).

Reference:

Grishin, A.M., Cherney, M., Anderson, D.H., Phanse, S., Babu, M., and Cygler, M. (2014). NleH defines a new family of bacterial effector kinases. **Structure** 22, 250-259.

Kannan, N., Taylor, S.S., Zhai, Y., Venter, J.C., and Manning, G. (2007). Structural and functional diversity of the microbial kinome. **PLoS Biol** 5, e17.

Lines 163-171 and Figs4b, regarding the transgenic plants, I wonder whether ectopic expression of XopC2 brought about any toxic effects on plant growth and development? To my eyes the leaves of IE17 are narrower than those of WT. How about those of OE-1 or of OE-10 plants? Whether expressing XopC2 affected stomatal development?

AUTHOR RESPONSE: To address the reviewer's concern, we investigated the growth phenotypes of the wild-type and *xopC2*-expressing transgenic plants. The transgenic lines IE-17, IE-37, OE-1 and OE-10 exhibited no significant difference from the wild-type plants in 4-week seedling height (Supplementary Fig. 3c, d), 4-month plant height (Supplementary Fig. 3e, f), leaf width (Supplementary Fig. 3g), and 100-grain weight (Supplementary Fig. 3i). These results indicate that ectopic expression of XopC2 has no toxic effect on the growth or development of the transgenic plants under natural growth conditions without pathogen infection or exogenous MeJA treatment. In Figure 2b and Supplementary Figure 4f, we showed that the stomatal conductance of the wild-type, OE-1, OE-10, DEX- or mock-treated IE-17 transgenic lines was not significantly different after spraying with MgCl₂, indicating that XopC2 expression does not affect stomatal development.

It seems to me that XopC2 expression affects the leaf color, so it will be informative if the authors can test the chlorophyll contents as well as carry out JA-induced leaf senescence assays given XopC2 could activate JA pathway.

AUTHOR RESPONSE: Thank you for your suggestion! As suggested, we quantified the chlorophyll content in the wild-type and *xopC2*-expressing transgenic plants without pathogen challenging. No significant difference in chlorophyll content was

detected between the wild-type and IE-17, IE-37, OE-1 and OE-10 transgenic lines (Supplementary Fig. 3h). Besides, we performed JA-induced leaf senescence assays. After treatment with exogenous MeJA, the detached leaves from the DEX-treated IE-17 and IE-37 transgenic plants showed an accelerated senescence as compared with the leaves from the wild-type and mock-treated IE-17 and IE-37 plants (Supplementary Fig. 5c). These data indicate that XopC2 expression activates JA signaling, but has no effect on chlorophyll content.

Also, I hope the authors can construct and observe the transgenic plants expressing XopC2 N396A, XopC2 D391A, respectively, along with those expressing XopC2. For example, whether the mutations could diminish bacterial disease severity? Fig2b the DEX-induced XopC2 expression compromises stomatal closure, whether this function of XopC2 is dependent to intact D391/N396?

AUTHOR RESPONSE: As the reviewer suggested, we generated the transgenic plants with DEX-induced expression of XopC2^{D391A} and XopC2^{N396A} (Supplementary Fig. 3b). We showed that these transgenic rice plants after DEX treatment did not have an enhanced disease susceptibility to the *ΔxopxC2* mutant strain (Fig. 2a). Besides, XopC2^{D391A} completely lose the ability to enhance expression of JA-responsive genes, such as *OsLOX2* and *OsJAZ8*, while XopC2^{N396A} largely compromised, but did not completely lose the ability to promote JA-induced gene expression (Fig. 3a, b).

In the revised manuscript, we also detected stomatal conductance of rice leaves in the *ΔxopxC2*-inoculated transgenic plants with expression of XopC2^{D391A} and XopC2^{N396A}. In contrast to XopC2 expression, XopC2^{D391A} expression did not compromise stomatal closure in the transgenic rice leaves induced by *ΔxopxC2* infection and XopC2^{N396A} expression slightly compromised stomatal closure in rice leaves induced by *ΔxopxC2* infection (Fig. 2b). These data indicate that XopC2 virulence function to compromise stomatal closure is dependent on intact D391/N396 and the kinase activity.

It will be helpful if the author can examine the gs at least at three time points to observe the possible effects of XopC2 on stomatal closure and or stomatal reopening.

AUTHOR RESPONSE: Thank you for your great suggestion! In the revised manuscript, we measured stomatal conductance at 0, 6, 12, 24, 48 and 72 hours post bacterial infection (Supplementary Fig. 4e). Stomatal conductance of rice leaves was gradually reduced till 24 h after bacterial infection. Compared with rice plants inoculated with *ΔxopxC2* and C-*ΔxopxC2*^{D391A} complemented strains, rice plants inoculated with the wild-type and C-*ΔxopxC2* complemented strains showed a significantly higher stomatal conductance at 24 hpi and thereafter. The results indicate that XopC2 suppresses stomatal closure, while XopC2^{D391A} loses this immunosuppressive ability.

Whether XopC2 can repress the expression of genes involved in SA biosynthesis and the accumulation of SA through JA pathway? Whether JA/SA content changes in response (are important) to XopC2?

AUTHOR RESPONSE: To address the reviewer concern, we measured the JA and SA contents in the wild-type and IE-17 transgenic rice plants after mock and DEX treatments (Supplementary Fig. 5d, e). The results showed that the JA and SA contents in the wild-type and IE-17 transgenic seedlings were unaltered regardless of mock and DEX treatments. Besides, we detected the expression of two SA biosynthesis genes *OsICS1* and *OsPAL1* in the wild-type and IE-17 transgenic plants (Supplementary Fig. 5f, g). It was demonstrated that DEX-induced expression of XopC2 in the IE-17 transgenic line did not alter the expression of *OsICS1* and *OsPAL1*. These data indicate that XopC2 does not alter the biosynthesis and accumulation of SA and JA.

Introduction section: the authors should introduce stomatal immunity properly to better establish the background of their study.

Author Response: Thanks for the suggestion! We have introduced stomatal immunity in the introduction section in the revised manuscript.

Abstract (line 50): I am not convinced that XopC2 “precisely” activates JA signaling.

Author Response: We have rewritten the abstract and deleted “precisely” from the abstract for cautiousness.

REVIEWERS' COMMENTS

Reviewer #1 (Remarks to the Author):

I have reviewed the revised manuscript and found that the revisions incorporated in the current version of the manuscript have addressed my concerns to my satisfaction.

Reviewer #2 (Remarks to the Author):

I have no more concerns. This revised report looks good to me.